# Unifying Understanding and Generation in Vision-Language Models: Advances, Challenges, and Opportunities

**Xiaocheng Lu[1,*], Ziyue Ma[1,2,*,†], Jie Zhang[1], Jian Liu[1], Song Guo[1]**
[1] *The Hong Kong University of Science and Technology*
[2] *Northwestern Polytechnical University*

Blue: revisions for Reviewer TCDu  Red: revisions for Reviewer r2Ft  Orange: revisions for Reviewer feLg

## Abstract

Significant advancements in vision-language models have predominantly followed two divergent trajectories: autoregressive architectures optimized for visual understanding and diffusion-based frameworks designed for high-fidelity generation. However, this separation hinders the development of truly versatile multimodal agents. Unifying these capabilities is a critical step toward Artificial General Intelligence, as recent findings suggest that effective understanding and generation can mutually reinforce each other. This survey provides a comprehensive overview of the emerging field of unified vision-language models and proposes a systematic taxonomy based on the core visual representation mechanism: *continuous* versus *discrete* visual tokens. For continuous visual tokens, we analyze how models bridge the semantic-visual gap by categorizing integration strategies into Serial Coupling, where LLMs act as planners, and Parallel Coupling, which enables bidirectional interaction. For discrete visual tokens, we contrast Autoregressive approaches that treat images as a foreign language against emerging Discrete Diffusion paradigms known for their global consistency and parallel decoding. Beyond architectural analysis, we provide a curated compilation of datasets and benchmarks essential for training and evaluation. Finally, we critically discuss open challenges such as tokenization trade-offs, training stability, and scalability, while outlining future directions for building seamless, omni-capable multimodal systems.

## 1 Introduction

The past few years have witnessed a transformative era in Artificial Intelligence, primarily fueled by the rapid advancement of Large Language Models (LLMs). Models such as LLaMA (Touvron et al., 2023), PanGu (Zeng et al., 2021), Qwen (Bai et al., 2023; Team et al., 2024; Yang et al., 2025a), and the GPT series (Brown et al., 2020; Achiam et al., 2023) have demonstrated unprecedented capabilities in understanding, reasoning, and text generation, fundamentally reshaping diverse applications. This success has naturally extended to multimodal domains, recognizing that true intelligence necessitates processing and synthesizing information beyond text. This evolution has given rise to powerful Vision-Language Models (VLMs) and advanced visual generation technologies, each progressing along distinct, yet highly impactful, trajectories.

In multimodal understanding, models primarily leverage autoregressive (AR) architectures (As shown in Figure 2a), building on the success of LLMs, to interpret diverse visual and textual inputs and generate discrete textual outputs that showcase deep semantic comprehension and reasoning capabilities. This approach benefits from AR models' strong capabilities in sequential modeling and nuanced reasoning. Conversely, visual generation has undergone its own rapid evolution. While initially explored by Generative Adversarial Networks (GANs) (Goodfellow et al., 2020), the field is now overwhelmingly dominated by diffusion models (Ho et al., 2020) (As shown in Figure 2b). Architectures like UNet (Ronneberger et al., 2015) and Diffusion

---
*Equal contribution.
†Ziyue Ma was an intern at HKUST.

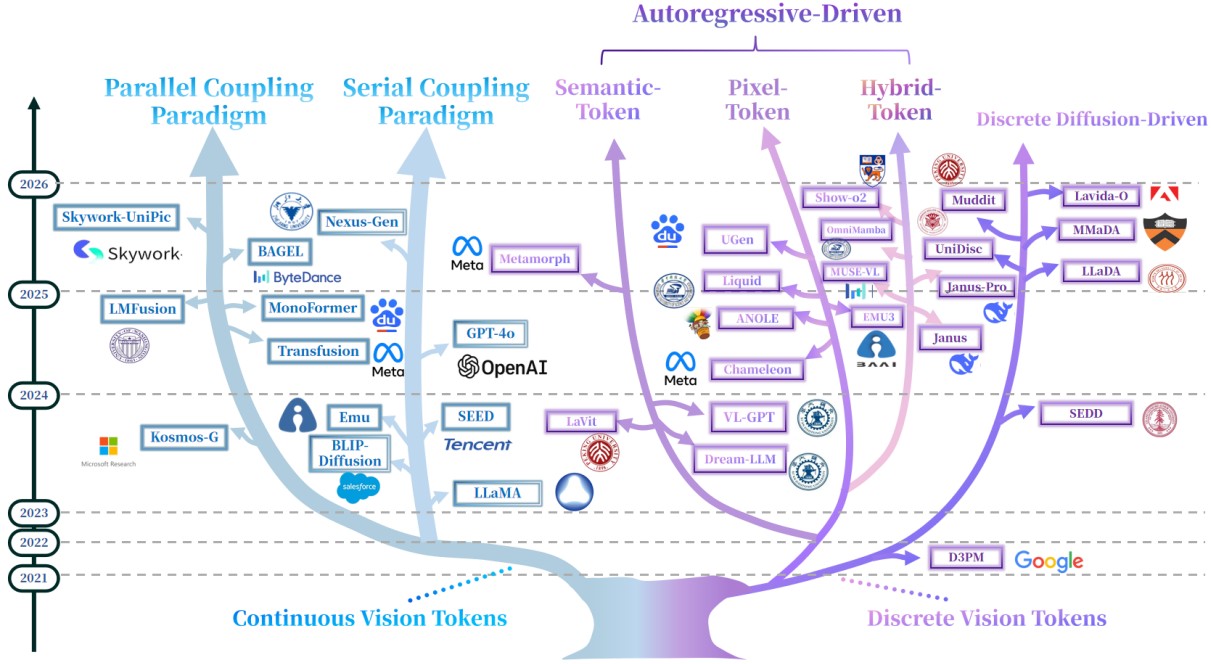

Figure 1: **Evolutionary Timeline (2021–2025) of Unified Understanding–Generation Models.** The diagram traces the convergence of multimodal AI from separate paradigms to unified frameworks. The branches distinguish models by their core visual representation: the blue stream represents *Continuous Vision Tokens* (prioritizing fidelity via diffusion), while the purple stream represent *Discrete Vision Tokens* (prioritizing alignment via AR or Discrete Diffusion). Key milestones highlight the rapid progression toward omni-capable agents.

Transformers (DiT) (Peebles & Xie, 2023), often conditioned by advanced text encoders such as CLIP (Radford et al., 2021) and T5 (Raffel et al., 2020), are proficient in synthesizing continuous, high-fidelity visual content from diverse prompts (e.g., Stable Diffusion series (Rombach et al., 2022; Podell et al., 2023)). However, pure diffusion-based generation models, despite their photorealism, fundamentally lack the deep world knowledge or true semantic understanding inherent in LLM-based systems. Their outputs are often faithful to the prompt's surface-level descriptions rather than reflecting nuanced conceptual comprehension. While some explorations have attempted to adapt LLM-inspired AR architectures for image generation (Sun et al., 2024; Tian et al., 2024), diffusion-based approaches currently maintain state-of-the-art performance in terms of visual quality.

The fundamental divergence in these underlying modeling paradigms—where AR is optimized for understanding and discrete text generation, versus Diffusion which excels in continuous visual synthesis but without intrinsic world knowledge—has historically posed a significant challenge for seamless unification into a singular framework. However, the vision of a unified model capable of both understanding and generating multimodal content holds immense potential. Such a framework could revolutionize how humans interact with AI, enabling complex visual reasoning guided by natural language, generating images from abstract instructions, and facilitating dynamic multimodal dialogues. The recent unveiling of GPT-4o (Hurst et al., 2024), demonstrating a singular model's ability to profoundly understand complex instructions and generate highly controllable multimodal content across these paradigms, vividly highlights this transformative potential and has galvanized research in this direction.

Designing such a truly unified framework presents substantial technical hurdles. A core challenge lies in the effective integration of AR's robust reasoning and text generation capabilities with Diffusion's generative quality and control. This often boils down to fundamental questions surrounding image representation and processing for unified models. For instance, how should visual information be tokenized for an AR backbone

when a continuous latent space is preferred for high-quality generation? Existing approaches vary, from employing VQ-VAE (Van Den Oord et al., 2017) or VQ-GAN (Esser et al., 2021) to create discrete visual tokens, to utilizing semantic encoders like EVA-CLIP (Sun et al., 2023a) or OpenAI-CLIP (Radford et al., 2021) that produce continuous embeddings. Furthermore, architectural design itself, encompassing purely autoregressive, diffusion-centric, or hybrid structures, presents various trade-offs that are still under active investigation.

To provide a comprehensive overview and foster future research in this rapidly converging and critical field (As shown in Figure 1), this survey systematically analyzes unified multimodal understanding and generation models. Our unique contribution lies in classifying existing unified models based on their core visual representation mechanisms: discrete visual tokens and continuous visual tokens. This classification enables a clearer comparison between models that emphasize the integration of textual understanding with discrete visual tokens (e.g., autoregressive models) and those that leverage continuous visual representations (e.g., diffusion models) for high-fidelity image generation. By highlighting the trade-offs and synergies between these approaches, we aim to uncover the strengths and limitations of each mechanism in the context of multimodal unification. Moreover, we discuss the implications of these design choices on model efficiency, scalability, and the ability to generate semantically rich and coherent visual content. Through this structured analysis, we aim to provide valuable insights and a roadmap for researchers seeking to advance the unification of understanding and generation within vision-language models, ultimately contributing to more powerful, efficient, and flexible multimodal AI systems.

**Taxonomy Preview** Unlike previous reviews that categorize models broadly by architecture, this work offers a specialized analysis grounded in the fundamental strategy of visual representation. As illustrated in Figure 1, we propose a hierarchical taxonomy that divides Unified Vision-Language Models into two primary categories, each with distinct sub-paradigms that offer unique trade-offs between flexibility, fidelity, and coherence.

**Unified Models with Continuous Vision Tokens.** These models interface LLMs with continuous latent representations, typically leveraging diffusion models to prioritize high-fidelity generation. We classify them based on how the reasoning (LLM) and generation (Diffusion) modules interact:

- **Serial Coupling Paradigm:** In this approach, the LLM functions as a "Semantic Planner," converting instructions into static embeddings that condition a separate visual generator (e.g., Emu, SEED). *Preview:* This paradigm excels in **modularity and scalability**, allowing the flexible combination of state-of-the-art LLMs and diffusion models. However, the unidirectional information flow creates a **"semantic-visual gap,"** as the LLM receives no feedback during the generation process, often limiting fine-grained alignment in complex spatial tasks.

- **Parallel Coupling Paradigm:** This emerging strategy fuses the LLM and visual generator within a unified attention span, enabling bidirectional interaction at each step (e.g., Transfusion, Skywork-UniPic). *Preview:* By allowing text and visual tokens to attend to each other, this paradigm achieves superior **semantic coherence and real-time interleaving**. The primary trade-off is higher **computational complexity** and the need for sophisticated training strategies to synchronize the discrete text space with the continuous latent space.

**Unified Models with Discrete Vision Tokens.** These models quantize visual data into discrete tokens (via VQ-VAE or VQ-GAN) that share a categorical vocabulary with text, effectively treating images as a "foreign language." We categorize them by their generative mechanism:

- **Autoregressive-Driven (AR):** Following the success of GPT, these models model visual tokens via next-token prediction (e.g., LLaViT, Show-o2). *Preview:* AR models benefit from a **unified training objective** that naturally aligns with LLMs, facilitating strong logical reasoning and instruction following. However, the unidirectional causal attention is suboptimal for visual data (which is naturally bidirectional), leading to issues like **error accumulation** and slow, serial inference speeds for high-resolution images.

- **Discrete Diffusion-Driven:** An innovative paradigm that applies diffusion processes directly to discrete tokens, predicting all tokens in parallel via iterative refinement (e.g., LLaDA, UniDisc). *Preview:* This approach combines the **alignment benefits** of a shared vocabulary with the **global coherence** of diffusion. It directly targets the serial bottleneck of AR models through **parallel decoding** and iterative self-correction, although the practical gain depends on tokenizer fidelity, refinement steps, and training stability. The challenge lies in the novelty of the method, which currently lacks the mature training recipes and stability of established AR or continuous diffusion frameworks.

**Definition and Scope of Unified Models**   We define a model as unified if it satisfies all of the following: (i) a single parameterization can both *understand* (text outputs) and *generate* (image outputs); (ii) the visual representation used for generation participates in the understanding pathway (shared tokens or a tightly coupled latent space), rather than being an external black-box; and (iii) cross-task transfer occurs within the same model without switching to a separate pipeline. *Non-examples* include loosely cascaded captioner → text-to-image pipelines connected only by prompts.

**Contributions**

- A representation-centric taxonomy: continuous vs. discrete visual tokens paired with interaction mechanisms (serial/parallel and AR/discrete-diffusion).

- Cross-axis meta-analysis and a practical roadmap: synthesized trends, failure modes, and concrete methods (hybrid/hierarchical tokenization, shared latent spaces, AR–Diffusion coupling schedules, multi-token/parallel decoding, unified data curricula and evaluation), with explicit evidence boundaries for heterogeneous results.

- Computational cost trade-offs: approximate comparisons of sequence length, attention complexity, decoding/refinement steps, tokenizer cost, memory, and latency across token types and coupling strategies.

- Ethics and safety: risks (misinformation/deepfakes, bias, environmental impact, multimodal jailbreaks) and mitigations, with explicit links to Section 8.

**Scope and objectives**   In comparison to existing surveys that broadly cover multimodal understanding or image generation, our work offers a unique and systematic analysis focusing specifically on unified multimodal understanding and generation models. Unlike previous reviews that might categorize models by architectural components (e.g., *autoregressive*, *diffusion*, *hybrid*), our distinctive approach is to classify these unified frameworks based on their core visual generation mechanisms: discrete visual generation and continuous visual generation. This novel categorization allows for a deeper exploration into how models bridge the gap between LLM-based understanding and diverse visual synthesis strategies. We meticulously detail the structural designs, visual representation strategies, and integration techniques with language models for each category. Furthermore, we provide a comprehensive compilation of relevant datasets and benchmarks, critically analyze key challenges including effective tokenization and cross-modal attention, and delineate promising future research directions. Our survey aims to provide a fresh perspective and a structured roadmap for researchers navigating the complexities of building truly integrated multimodal AI systems.

**Organization**   The remainder of this survey is organized as follows. Chapter 2 lays the groundwork by introducing the fundamental concepts and recent advances in Large Language Models (LLMs), multimodal understanding, and visual generation, setting the stage for unified models. Chapter 3 focuses on unified models that use Continuous Visual Generation, detailing their unique characteristics and technical innovations. Following this, in Chapter 4, we dive into unified models that employ discrete visual generation mechanisms, analyzing their architectural designs, visual representation strategies, and integration with language models. Chapter 5 provides a comprehensive compilation of relevant datasets and benchmarks crucial to the training and evaluation of these sophisticated unified models. Finally, Chapters 6–8 discuss challenges, opportunities, summary directions, and ethics/safety considerations, aiming to inspire further advancements in unified multimodal AI.

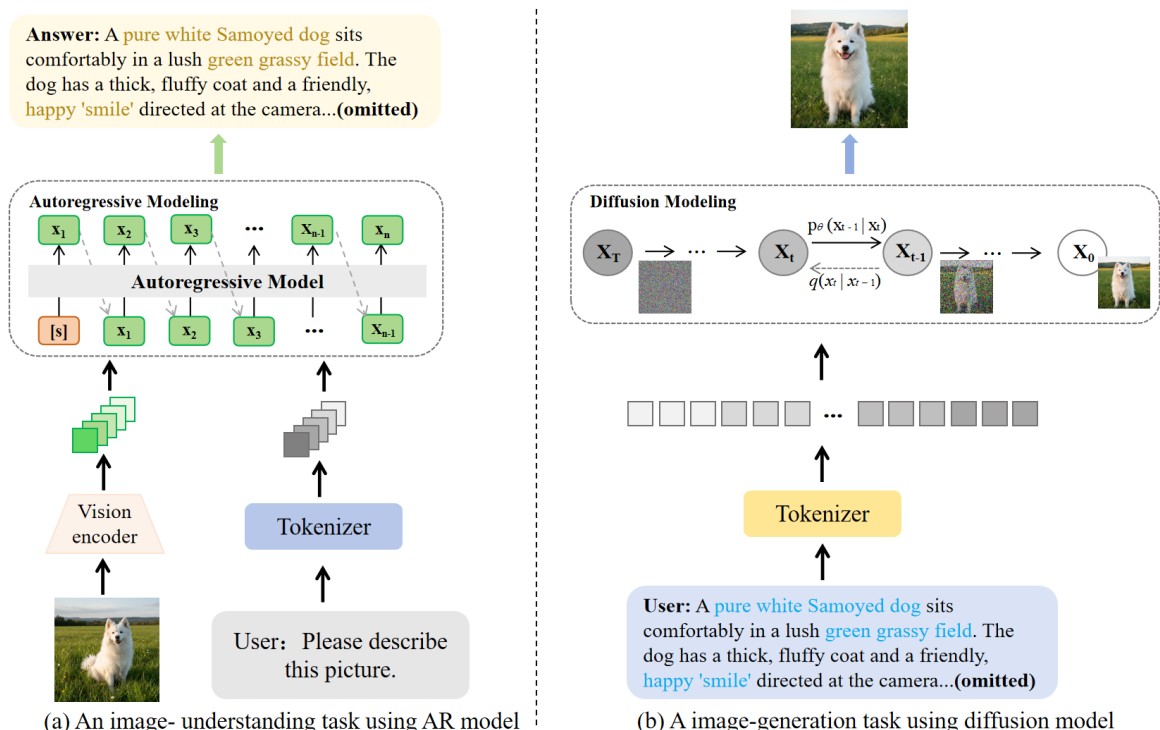

**Answer:** A pure white Samoyed dog sits comfortably in a lush green grassy field. The dog has a thick, fluffy coat and a friendly, happy 'smile' directed at the camera...**(omitted)**

**Autoregressive Modeling**

$x_1$ $x_2$ $x_3$ ··· $x_{n-1}$ $x_n$

**Autoregressive Model**

[s] $x_1$ $x_2$ $x_3$ ··· $x_{n-1}$

Vision encoder

Tokenizer

User： Please describe this picture.

(a) An image- understanding task using AR model

**Diffusion Modeling**

$p_\theta(x_{t-1}|x_t)$

$X_T$ → ··· → $X_t$ → $X_{t-1}$ → ··· → $X_0$

$q(x_t|x_{t-1})$

Tokenizer

**User:** A pure white Samoyed dog sits comfortably in a lush green grassy field. The dog has a thick, fluffy coat and a friendly, happy 'smile' directed at the camera...**(omitted)**

(b) A image-generation task using diffusion model

Figure 2: The illustration of Autoregressive-based Semantic Understanding and Diffusion-based Visual Generation.(a) Autoregressive (AR) modeling for image understanding, where visual features extracted by a vision encoder are aligned with textual tokens and decoded sequentially to generate semantic descriptions. (b) Diffusion-based modeling for image generation, where textual tokens produced by a tokenizer condition an iterative denoising process to synthesize visual content from noise.

## 2   Background

This chapter provides an overview of the background behind Vision-Language Models (VLMs), focusing on their unified capabilities in both understanding and generation. We start by reviewing Large Language Models (LLMs), which have revolutionized NLP with their generative and reasoning abilities. We then explore visual understanding methods, detailing how visual information is encoded and integrated with LLMs. Finally, we discuss text-to-image generation models, which, combined with visual understanding, lay the foundation for unified multimodal AI.

### 2.1   Large Language Models

The Transformer architecture's rise, initially enabling discriminative (e.g., BERT (Devlin et al., 2019)) and generative (e.g., GPT-2 (Radford et al., 2019)) NLP tasks, culminated in a paradigm shift with generative large language models (LLMs). Driven by exponential scaling and enhanced by techniques like instruction tuning (Shengyu et al., 2023) and RLHF (Ouyang et al., 2022), LLMs unified diverse NLP applications into sophisticated instruction-following and reasoning frameworks (Wei et al., 2022). Their emergent abilities and tool integration have transformed them into versatile agents, establishing LLMs as the crucial foundation for unified multimodal intelligence.

### 2.2   Vision Representations Methods

Vision representations play a critical role in bridging visual inputs with language models in unified vision-language systems. These representations can be learned using both supervised and unsupervised (or self-

supervised) methods, each offering distinct advantages depending on the task and available data. In this section, we provide an overview of the most widely used approaches for visual representations, categorized into supervised and unsupervised learning.

**Supervised Learning-Based Methods.** In supervised learning, vision representations are learned from labeled datasets, with models trained to map images to feature vectors aligned with specific labels. CNNs and Vision Transformers (ViT) are commonly used as backbones. ResNet (He et al., 2016) introduced deep residual networks to facilitate training of very deep architectures. DenseNet (Huang et al., 2017) connects every layer to each other, improving compactness and reducing overfitting. A significant advancement is the CLIP model (Radford et al., 2021), which uses contrastive learning to align visual and textual representations, enabling zero-shot classification, image captioning, and visual question answering. Additionally, the Segment Anything Model (SAM) (Kirillov et al., 2023) has revolutionized segmentation by providing high-quality, pixel-level segmentation masks across diverse visual inputs, making it highly effective for tasks requiring precise localization.

**Self-Supervised Learning-Based Methods.** Unsupervised and self-supervised learning methods learn visual representations without relying on labeled data, instead exploiting the inherent structure of the data. Self-supervised learning (SSL) has gained popularity for learning robust features without annotations. The DINO model (Caron et al.; Oquab et al., 2023; Siméoni et al., 2025) contrasts augmentations of the same image to generate consistent, transferable representations. Similarly, VAE (Variational Autoencoders) (Kingma & Welling, 2013) learns compact latent variables to model the underlying structure of images, excelling in generative tasks. These methods, such as DINO and VAE, have proven highly effective in producing generalizable representations for tasks like object detection and recognition, demonstrating the power of unsupervised learning even without large labeled datasets.

## 2.3 Visual Understanding Methods

Multimodal understanding models extend LLM reasoning to visual information, enabling tasks like VQA and multimodal dialogue. The core challenge is bridging continuous visual signals with the LLM's discrete token space. The first technical route is **visual encoding**. While early methods used pre-trained encoders (e.g., ViT (Dosovitskiy, 2020)) with simple projection layers (Zhu et al., 2023a), more sophisticated approaches use querying transformers (e.g., Q-Former (Li et al., 2023c)) or gated cross-attention (Alayrac et al., 2022) to selectively distill relevant visual features into soft tokens compatible with the LLM.

The second route is **information fusion**. Initial dual-encoder architectures (Lu et al., 2019; Radford et al., 2021) focused on aligning latent spaces. The modern, more effective approach is LLM-centric: visual tokens are fed directly into the LLM backbone, allowing for deep cross-modal reasoning via the LLM's internal self-attention mechanisms. This is often initialized by multi-stage alignment training (Chen et al., 2024) and can be enhanced by advanced architectures like Mixture-of-Experts (MoE) (Wu et al., 2024b) to improve semantic coherence.

## 2.4 Visual Generation Methods

Visual generation models, critical for unified systems, synthesize visual content from text. This field is dominated by two paradigms: diffusion models (DMs) and autoregressive (AR) models. DMs excel in quality and diversity, while AR models offer strong sequential consistency.

DMs model generation as an iterative denoising process (Ho et al., 2020). The pivotal shift was Latent Diffusion Models (LDMs) (Rombach et al., 2022), which operate in a pre-trained VAE's compact latent space, enhancing efficiency. More recently, Diffusion Transformers (DiT) (Peebles & Xie, 2023) treat image patches as sequences. Textual conditioning is incorporated via encoders like CLIP (Radford et al., 2021) or T5 (Raffel et al., 2020), and increasingly, LLMs are used for richer, more aligned conditioning (Zhang et al., 2024; Wang et al., 2025a).

AR models frame image generation as a sequential prediction task (Van Den Oord et al., 2016). The core challenge is discretizing visual data. While early models predicted pixels (e.g., PixelCNN (Van den Oord et al., 2016)), the breakthrough came from token-based models using vector quantization (e.g., VQGAN (Esser

et al., 2021)). A Transformer decoder then predicts these visual tokens sequentially. Recent work improves efficiency via multi-token prediction (Pang et al., 2024), coarse-to-fine strategies (e.g., VAR (Tian et al., 2024)), or control mechanisms (e.g., ControlAR (Li et al., 2024f)) for precise editing.

# 3 Unified models with continuous vision tokens

This chapter provides a comprehensive overview of **continuous tokenizers** and their corresponding **generation mechanisms**, focusing on continuous diffusion and autoregressive (AR) frameworks. We begin by introducing the theoretical foundation of continuous tokenization, followed by detailed diffusion and autoregressive formulations. Subsequent sections discuss how these mechanisms are integrated within unified multimodal architectures through serial and parallel coupling paradigms.

## 3.1 Preliminaries

### 3.1.1 Continuous Tokenizers: Foundations and Formulation

A **continuous tokenizer** bridges high-dimensional visual inputs and compact, semantically meaningful latent representations that generative models can directly operate on. Unlike discrete quantization methods, continuous tokenizers learn smooth latent manifolds, where each visual feature is represented as a differentiable vector. This continuous design preserves gradient flow and fine-grained details, making it suitable for end-to-end training in diffusion and autoregressive systems.

The *Variational Autoencoder (VAE)* provides the theoretical foundation for continuous tokenization. Given an input image $x$, the encoder $q_\phi(z|x)$ maps it to a latent variable $z$, while the decoder $p_\theta(x|z)$ reconstructs $x$. Training maximizes the Evidence Lower Bound (ELBO):

$$\mathcal{L}_{\text{ELBO}}(\theta, \phi; x) = \mathbb{E}_{q_\phi(z|x)}[\log p_\theta(x|z)] - \text{KL}(q_\phi(z|x) \,\|\, p(z)), \tag{1}$$

where $p(z)$ is typically a standard Gaussian prior. Compared to discrete tokenizers such as VQ-VAE, VAEs maintain continuous, differentiable representations, allowing gradient propagation through the latent space.

### 3.1.2 Diffusion Generation

Diffusion models define a forward noising process and a learned reverse denoising process. Given a clean sample $x_0$, Gaussian noise is progressively added:

$$q(x_t|x_{t-1}) = \mathcal{N}(x_t; \sqrt{1 - \beta_t}\, x_{t-1}, \beta_t I),$$

which can be expressed as:

$$x_t = \sqrt{\bar{\alpha}_t}\, x_0 + \sqrt{1 - \bar{\alpha}_t}\, \epsilon, \quad \epsilon \sim \mathcal{N}(0, I),$$

where $\bar{\alpha}_t = \prod_{s=1}^{t}(1 - \beta_s)$. A neural predictor $\epsilon_\theta(x_t, t)$ learns to reverse this process by minimizing the simplified denoising loss (Ho et al., 2020):

$$\mathcal{L}_{\text{simple}}(\theta) = \mathbb{E}_{x_0, \epsilon, t}\big[\|\epsilon - \epsilon_\theta(x_t, t)\|^2\big].$$

**Latent Diffusion Models (LDM)** (Rombach et al., 2022) improve efficiency by operating in a learned latent space instead of pixel space. An encoder $E$ compresses $x$ into $z = E(x)$, diffusion is applied on $z$, and a decoder $D$ reconstructs $x_0 \approx D(z_0)$. Conditioning signals (e.g., text or depth) can be injected via cross-attention, leading to controllable high-fidelity generation at reduced cost. Extensions such as FLUX further enhance training efficiency through rectified flow objectives and multi-scale latent processing.

### 3.1.3 Autoregressive Continuous Generation

Traditional autoregressive (AR) models factorize discrete sequences:

$$p(s_{1:N}) = \prod_{t=1}^{N} p(s_t \mid s_{<t}),$$

but for continuous latent embeddings $z_t \in \mathbb{R}^d$, direct regression may fail to capture multimodal distributions. **Continuous AR** approaches retain AR dependencies while replacing discrete categorical modeling with diffusion-based conditional generation.

Specifically, for each token $z_t$, we define a noisy version:

$$z_t^{(\tau)} = \sqrt{\bar{\alpha}_\tau}\, z_t + \sqrt{1 - \bar{\alpha}_\tau}\, \epsilon, \quad \epsilon \sim \mathcal{N}(0, I),$$

and train a denoiser conditioned on previous tokens $z_{<t}$:

$$\mathcal{L}_{\text{token}}(\theta) = \mathbb{E}_{z_t, \epsilon, \tau}\Big[\|\epsilon - \epsilon_\theta(z_t^{(\tau)}, \tau; \text{context} = z_{<t})\|^2\Big].$$

This formulation combines diffusion robustness with AR's sequential modeling power, as demonstrated in MAR (Li et al., 2024e), Fluid (Fan et al., 2024), and Emu (Sun et al., 2023b).

## 3.2 Coupling Strategies for Continuous Unified Models

The integration of large language models (LLMs) with continuous visual generators defines two main paradigms: **serial coupling** and **parallel coupling** (as shown in Figure 3). They differ by how reasoning and generation interact—either sequentially or concurrently—forming a spectrum from modular to deeply unified multimodal architectures.

To avoid conflating unified models with ordinary pipelines, we use the following operational distinction. A *serially coupled* system is still relevant to unified modeling when the LLM output is not merely a natural-language prompt, but a structured visual condition, latent plan, or learned connector representation that is optimized for the downstream generator. A purely chained captioner–prompt–generator pipeline is therefore a weak or non-unified boundary case. A *parallel-coupled* system, in contrast, exposes language and visual latent states to each other inside shared layers, shared attention, or jointly trained expert routes; its defining property is feedback during representation formation or denoising, not simply the presence of both an LLM and a diffusion model. This distinction clarifies why serial systems mainly test modularity and deployment efficiency, whereas parallel systems test whether reasoning and generation can be co-optimized within a common computational graph.

### 3.2.1 Serial Coupling Paradigm

In the **serial coupling** paradigm, the large language model (LLM) functions as a *semantic planner* that transforms high-level textual descriptions into structured conditioning signals for continuous visual generators. The generation pipeline follows a strictly modular and sequential formulation: the LLM first performs semantic reasoning and produces a latent condition $c$, after which a diffusion or flow-based generator consumes this representation to synthesize image latents. This design inherits the strengths of modular multimodal systems, enabling each component—textual reasoning and visual synthesis—to be independently optimized while leveraging powerful pretrained models across modalities.

In contemporary implementations, the LLM interprets and restructures user instructions into dense semantic embeddings, which are injected into latent diffusion models through cross-attention, adapter routing, or learnable query connectors. Systems such as BLIP-Diffusion (Li et al., 2023c) and Emu (Sun et al., 2023b) exemplify this approach: BLIP-Diffusion uses BLIP-2 as a multimodal planner to extract semantic visual features, while Emu introduces a hybrid autoregressive–diffusion mechanism where the LLM predicts coarse latent representations that are subsequently refined by a diffusion model. The SEED family (Ge et al.; 2023; 2024) extends this paradigm to industrial-scale pipelines by coupling reasoning from Vicuna or LLaMA with latent diffusion backbones such as SDXL or SANA, mediated by lightweight connectors that preserve scalability. Large commercial systems—including Kosmos-G (Pan et al., 2023), Nexus-Gen (Zhang et al., 2025b), GPT-4o, and Gemini-1.5—also follow serial pipelines that map textual reasoning into structured latent conditions for continuous generation.

Despite its practicality and modularity, serial coupling introduces an inherent *semantic–visual gap*. Because the generation module does not provide gradient-level feedback to the LLM, mismatches may arise between

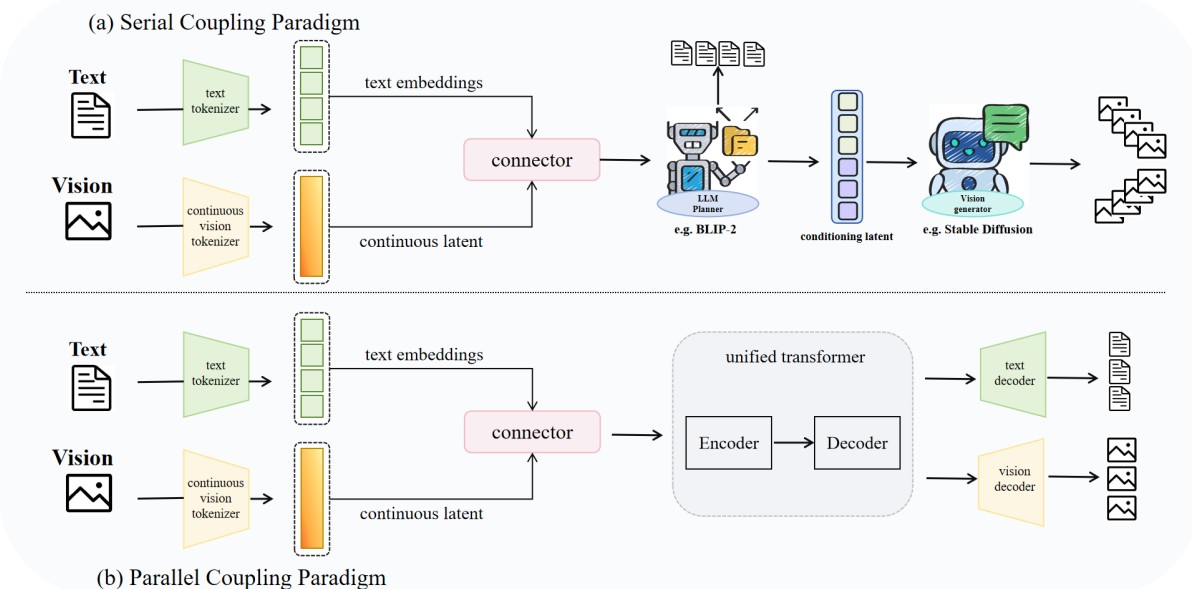

Figure 3: The illustration of serial coupling and parallel coupling. (a) Serial coupling, where multimodal inputs are first processed by a reasoning-oriented language model to produce intermediate semantic representations, which are subsequently used to condition a separate visual generation module. (b) Parallel coupling, where multimodal inputs are jointly processed within a unified transformer architecture, enabling simultaneous reasoning and visual generation through shared representations and coordinated decoding.

the semantic planner and the visual decoder, particularly for complex layouts or fine-grained attributes. Moreover, the strictly unidirectional pipeline prevents dynamic interplay between reasoning and generation during the denoising process, limiting cross-modal coherence. While serial coupling remains favored for its flexibility, scalability, and ease of integration with pretrained LLMs, it stops short of achieving fully unified multimodal reasoning–generation within a single model architecture.

### 3.2.2 Parallel Coupling Paradigm

The **parallel coupling** paradigm differs fundamentally from the sequential nature of serial coupling by enabling *synchronous, bidirectional interaction* between linguistic reasoning and continuous visual generation. Instead of treating the LLM as an offline planner, parallel coupling integrates visual denoising and text reasoning within a shared or partially shared transformer architecture, allowing the two modalities to co-evolve during generation. In this formulation, text tokens typically retain causal attention to preserve autoregressive language modeling, while visual latent tokens employ bidirectional attention to accommodate diffusion or flow dynamics. The shared attention space ensures that linguistic semantics and visual structure continuously influence each other, thereby achieving deeper multimodal unification.

This paradigm is exemplified by models such as Transfusion (Zhou et al., 2024) and MonoFormer (Zhao et al., 2024), which adopt unified transformer backbones with distinct attention masks for language and vision. In Transfusion, autoregressive text generation and latent diffusion denoising are fused within a single architecture: language tokens are processed with causal masking whereas image tokens use full bidirectional attention, with cross-attention pathways enabling mutual guidance. MonoFormer follows a similar design but simplifies the multimodal integration through shared layers and structured attention routing, providing efficient interleaving of text reasoning and latent-space image generation. These models demonstrate how integrating text and vision streams at every generation step provides stronger semantic alignment than a serial pipeline could achieve.

LMFusion (Shi et al., 2024) extends the parallel paradigm by adapting pretrained LLMs to serve as the backbone for multimodal interaction. Lightweight visual modules and shared attention layers are introduced to merge visual features into the frozen language backbone, enabling the model to leverage the stability of LLM parameters while achieving effective bidirectional fusion between reasoning and perception. BAGEL (Deng et al., 2025) generalizes this idea via a Mixture-of-Transformers (MoT) framework. It separates modality- and task-specific attention routes while maintaining a unified architecture, allowing flexible yet coherent interactions across tasks such as image generation, editing, captioning, and dialogue. Although powerful, these architectures require substantial computation due to the need for cross-modal synchronization and large unified transformer backbones.

A notable recent advancement within the parallel coupling landscape is Skywork-UniPic (Wang et al., 2025b), which demonstrates that unified autoregressive modeling can support visual understanding, generation, and editing within a single architecture. UniPic employs a decoupled but jointly trained visual encoding pathway—a Masked Autoregressive (MAR) encoder-decoder for generation and a SigLIP2 encoder for perception—while sharing a single language backbone across tasks. This design allows text reasoning and visual latent evolution to take place concurrently, embodying the essence of parallel coupling: language tokens guide visual synthesis at each step, while image latent tokens reciprocally inform linguistic processing. UniPic highlights that continuous visual generation and language modeling can be co-trained and co-executed without handcrafted connectors, reinforcing the shift toward tightly unified multimodal systems.

Overall, parallel coupling offers significantly enhanced cross-modal coherence, enabling functionalities such as real-time editing, interactive visual dialogue, instruction-driven fine-grained manipulation, and open-ended multimodal reasoning. However, such tightly integrated architectures come with increased training and inference costs, complex token routing requirements, and higher sensitivity to optimization dynamics. Despite these challenges, parallel coupling represents a critical step toward fully unified multimodal foundation models capable of seamless joint reasoning and generation.

### 3.2.3   Analysis and Outlook

Serial coupling prioritizes scalability and modularity, while parallel coupling maximizes coherence through shared representation learning. Emerging hybrid models combine both—serial pipelines with partial parallel feedback—to balance efficiency and multimodal alignment. Future directions point toward *adaptive coupling*, dynamically controlling the degree of inter-modality interaction based on task context, progressing toward fully unified multimodal foundation models.

### 3.3   Recent Directions: Semantic Features, Continuous AR, and Pixel-Space Unification

The most recent generation of unified models suggests that the "continuous-token" family is itself splitting into several finer-grained routes. Rather than treating continuous visual representations as synonymous with VAE latents, newer systems explore semantic feature spaces, autoregressive continuous tokens, high-resolution visual encoders, and even raw-pixel embeddings. This refinement is important because the central bottleneck in unified modeling is no longer merely whether the visual representation is continuous or discrete, but whether it preserves the right mixture of semantics, spatial detail, reconstruction fidelity, and language-model compatibility.

**Continuous autoregressive visual tokens.** UniFluid (Fan et al., 2025) studies a pure autoregressive framework in which text outputs remain discrete tokens while image outputs are continuous visual tokens predicted token by token. It uses a Gemma-family decoder-only backbone, a continuous tokenizer for image generation, a SigLIP encoder for visual understanding, and modality-specific prediction heads. Its main empirical message is that visual generation and understanding can improve each other under a carefully tuned loss balance, but the trade-off is sensitive: increasing the text loss weight improves understanding while quickly degrading generation quality. UniFluid also highlights that stronger language-model initialization and random-order image-token training are crucial for high-fidelity continuous-token generation.

**Semantic-feature generation.** BLIP3-o (Chen et al., 2025a) provides another route by generating CLIP image features rather than directly predicting VAE latents. This design shifts the generative target toward

Table 1: Comparison between the **Serial Coupling** and **Parallel Coupling** paradigms for unified continuous visual models.

| Aspect | Serial Coupling | Parallel Coupling |
|---|---|---|
| **Core Philosophy** | LLM acts as a *semantic planner*; reasoning and generation occur sequentially. | Text reasoning and visual denoising occur *synchronously* within shared or partially shared modules. |
| **Information Flow** | Unidirectional: LLM → visual generator. | Bidirectional: language and vision tokens interact at every step of generation. |
| **Representation Interaction** | Loose coupling; cross-attention or adapters provide semantic constraints. | Tight coupling; shared transformer blocks or unified attention layers align modalities. |
| **Generative Process** | Visual generation is conditioned on static semantic embeddings. | Visual denoising dynamically conditions on evolving language states, and vice versa. |
| **Advantages** | Modular, scalable, easy to integrate with pretrained LLMs; efficient for large-scale deployment. | Strong cross-modal coherence; supports interactive editing, multimodal reasoning, and fine-grained control. |
| **Limitations** | Semantic–visual gap; no gradient feedback from image to LLM; limited real-time alignment. | High computational cost; complex attention scheduling; difficult optimization due to tight coupling. |
| **Representative Systems** | BLIP-Diffusion (Li et al., 2023c), Emu (Sun et al., 2023b), SEED (Ge et al.; 2023; 2024), Kosmos-G (Pan et al., 2023), Nexus-Gen (Zhang et al., 2025b). | Transfusion (Zhou et al., 2024), Mono-Former (Zhao et al., 2024), LMFusion (Shi et al., 2024), BAGEL (Deng et al., 2025), UniPic (Wang et al., 2025b). |
| **Suitable Use Cases** | High-throughput generation, instruction-to-image pipelines, modular system design. | Unified multimodal modeling, real-time multimodal tasks, understanding–generation joint optimization. |

a semantically rich visual space, making it easier to couple image generation with language reasoning and visual understanding. Compared with VAE-latent generation, CLIP-feature generation is less tied to low-level reconstruction and more aligned with high-level concepts, but it still requires a downstream visual decoder or generator to recover pixels. This makes BLIP3-o a useful midpoint between purely semantic understanding encoders and reconstruction-oriented image tokenizers.

**High-resolution semantic encoders.** UniWorld (Lin et al., 2025a) argues that unified models can benefit from stronger high-resolution semantic encoders rather than removing encoders entirely. In contrast to approaches that rely on low-resolution CLIP-style features or separate generation-only VAEs, UniWorld uses semantic features to improve perception, manipulation, and generation with comparatively efficient data usage. This line of work suggests a pragmatic alternative to fully end-to-end pixel modeling: retain a powerful semantic encoder, but make it sufficiently high-resolution and generation-aware to reduce the understanding–generation mismatch.

**Pixel-space and encoder-free modeling.** Tuna-2 (Liu et al., 2026) pushes the opposite direction by progressively removing visual modules. Starting from a representation-encoder variant (Tuna-R), it eliminates the VAE and finally discards the pretrained visual representation encoder, replacing it with simple patch embeddings over raw pixels. Image generation is performed by pixel-space flow matching, while understanding uses the same transformer decoder over text and pixel-derived image tokens. Its controlled comparison shows a useful pattern: pretrained visual encoders accelerate early training and slightly help generation, but an encoder-free pixel-space model can surpass encoder-based variants on fine-grained visual understanding after sufficient pretraining. This result weakens the assumption that CLIP/SigLIP-like encoders are necessary for unified multimodal understanding and generation.

**Engineering-oriented unified generation.** Recent systems such as OmniGen (Xiao et al., 2024a), OmniGen2 (Wu et al., 2025b), and Mogao (Liao et al., 2025) focus on unifying practical generation modes: text-to-image generation, image editing, subject-driven generation, in-context generation, and interleaved text–image outputs. These models are not always "unified" in the strictest sense of sharing every representation across understanding and generation, but they are important because they expose the deployment pressure points that academic unified models must eventually solve: controllability, identity consistency, multi-image context, arbitrary interleaving, latency, and editable outputs. Mogao is particularly relevant for

survey taxonomy because it treats interleaved multimodal generation as a first-class target rather than an auxiliary demonstration.

Taken together, these developments reveal two competing but complementary tendencies. One direction strengthens visual priors through semantic encoders or CLIP-feature targets, improving data efficiency and controllability. The other direction removes hand-crafted visual modules and moves toward raw pixels and end-to-end learning. A useful future taxonomy should therefore distinguish not only discrete versus continuous tokens, but also *reconstruction-oriented latents*, *semantic visual features*, *autoregressive continuous vectors*, and *pixel-space embeddings*.

### 3.4 Challenges for Continuous Unified Models

Continuous visual representations bring advantages such as high perceptual fidelity, differentiability, and natural compatibility with diffusion and flow models. Nonetheless, they also present challenges: weak interpretability, substantial computational cost, and non-trivial semantic alignment between continuous vision and discrete language spaces. Addressing these limitations remains central to advancing continuous unified multimodal systems.

## 4 Unified models with discrete vision tokens

Unified multimodal models employing discrete visual generation represent a powerful paradigm where both understanding and generation operate within a shared, tokenized conceptual space. Unlike continuous generation methods that manipulate latent vectors, these models treat visual content—much like text—as sequences of discrete tokens. This approach naturally aligns with the sequence-to-sequence nature of Large Language Models (LLMs), allowing the entire understanding-to-generation pipeline to be executed by a single, often Transformer-based, backbone.

The primary motivation for adopting discrete visual representations lies in their ability to leverage the powerful capabilities of LLMs, which have been extensively pre-trained on vast amounts of textual data. By transforming visual content into discrete tokens, these models enable the integration of vision tasks with the well-established techniques used for language tasks, facilitating the unification of understanding and generation. This tokenization simplifies the bridging of the semantic gap between language and vision, as it allows the use of a shared framework that processes both modalities in a similar manner.

Furthermore, discrete visual representations bring significant advantages in terms of **efficiency** and **coherence** with language. By converting visual content into a set of discrete tokens, visual information is structured in a way that aligns more directly with the discrete nature of text. This structural alignment allows vision tasks to be integrated seamlessly into a unified framework, where both visual and language models operate in a similar tokenized space. As a result, discrete models facilitate **cross-modal coherence**, making it easier for the system to handle both vision and language tasks simultaneously, with the visual content serving as tokens that can be processed in parallel with text. This approach enables **modularization** of the vision and language components. Each modality can be independently optimized and improved, while still ensuring that they can interact effectively within the same architecture. Additionally, discrete representations are often more **computationally efficient** compared to continuous models. The discrete tokens are easier to process and store, which reduces the computational burden, especially in large-scale multimodal tasks. By simplifying the representation of visual data, discrete models enable efficient scaling to high-dimensional visual and textual data without compromising the **alignment** between the two modalities.

This chapter first explores the critical process of converting continuous visual signals into discrete representations. Subsequently, it delves into the two primary methodologies for generating these discrete visual tokens within unified frameworks (as shown in Figure 4): autoregressive-driven generation and discrete diffusion-driven generation.

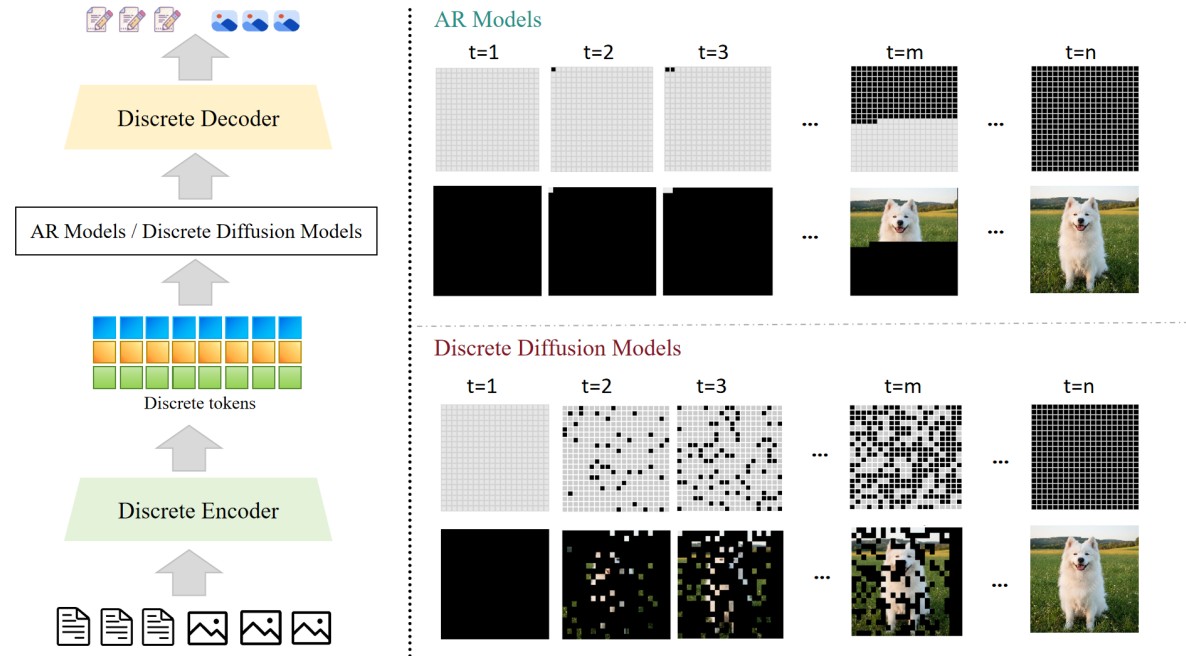

Figure 4: Comparison of Autoregressive and Diffusion Paradigms for unified models with discrete vision tokens.

## 4.1 Visual Vector Quantization

A unified multimodal framework hinges on the ability to represent continuous visual signals—such as images, videos, or spatial-temporal scenes—as discrete sequences of tokens. This transformation enables vision data to share the same representational format as language, allowing both understanding and generation to be handled by a single sequence model. The challenge lies in designing discrete representations that are compact, expressive, and semantically aligned across modalities.

### 4.1.1 Preliminary on Vector Quantization

Most discrete visual representations stem from the principle of vector quantization (VQ), first introduced by *VQ-VAE* (Van Den Oord et al., 2017). In these frameworks, an encoder $\text{Enc}(\cdot)$ maps the input $x$ into a latent feature map $z = \text{Enc}(x)$, which is then discretized by replacing each latent vector with its nearest entry from a learned codebook $\mathcal{C} = c_1, \ldots, c_M$:

$$Q(z_i) = \arg\min_{c_j \in \mathcal{C}} \|z_i - c_j\|_2.$$

A decoder $\text{Dec}(\cdot)$ reconstructs $\hat{x} = \text{Dec}(Q(z))$, and the entire system is trained to minimize reconstruction and commitment losses. This process produces discrete code indices that serve as "visual tokens," analogous to word tokens in NLP.

The key challenge in VQ is managing the trade-off between compression and fidelity. *VQ-GAN* (Esser et al., 2021), for instance, incorporates adversarial and perceptual losses to recover high-frequency details. Subsequent refinements have focused on common failure modes: hierarchical latents (e.g., VQ-VAE-2 (Razavi et al., 2019)) capture multi-scale features, while various regularization techniques (e.g., Reg-VQ (Zhang et al., 2023a), HC-VQ (Volkov, 2022)) and re-initialization strategies (e.g., HQA (Williams et al., 2020), CVQ-VAE (Zheng & Vedaldi, 2023)) are used to prevent codebook collapse. Soft-assignment methods (e.g., SQ-VAE (Takida et al., 2022), SCQ (Gautam et al., 2023)) have also been explored to create richer, less-constrained representations.

### 4.1.2 Advanced Variants of Vector Quantization

**Residual Vector Quantization (RVQ)** decomposes latent vectors into multiple residual stages, progressively refining quantized codes by quantizing the residual error from previous stages. This hierarchical approach allows for multi-scale representation, capturing both coarse and fine details (Barnes et al., 1996; Chen et al., 2010). Variants primarily focus on optimizing the stages, such as using hierarchical dependency (SQ (Martinez et al., 2014)), joint optimization (ErVQ (Ai et al., 2017)), or subspace clustering (IRVQ (Liu et al., 2015)). Recent approaches like QINCo (Huijben et al., 2024) use neural networks to generate the step-specific codebooks.

**Product Quantization (PQ)** partitions the latent space into independent subspaces, each with its own sub-codebook, greatly expanding the effective vocabulary size without linear parameter growth. This method is particularly efficient for high-dimensional visual data (Jegou et al., 2010; Matsui et al., 2018). Extensions focus on optimizing the subspace decomposition (OPQ (Ge et al., 2013), LOPQ (Kalantidis & Avrithis, 2014)), enabling online updates (Online PQ (Xu et al., 2018)), or making the process differentiable (DPQ (Klein & Wolf, 2019), DOPQ (Lu et al., 2023b)).

**Additive Vector Quantization (AQ)** represents vectors as sums of codewords to promote sparsity and efficiency (Babenko & Lempitsky, 2014). Variants include LSQ (Martinez et al., 2016), which uses iterated local search for encoding. Recent developments, such as **Finite Scalar Quantization (FSQ)** (Mentzer et al., 2023) and **Lookup-Free Quantization (LFQ)** (Yu et al., 2023a), avoid explicit codebooks by using rounding or sign-based decompositions. These approaches (e.g., LFQ as used in MAGVIT-v2 (Yu et al., 2023a)) mitigate codebook collapse and improve training stability, facilitating end-to-end optimization.

### 4.1.3 Challenges and Future Directions

**Challenges.** One primary challenge is the *codebook collapse* problem, where certain visual tokens are never utilized during training, leading to inefficient representations. This is particularly problematic when attempting to align visual and language tokens in a unified space. Furthermore, the *quantization error* introduced during the conversion of continuous visual signals to discrete tokens can accumulate, affecting both understanding and generation tasks. This issue is especially prominent in tasks that require high fidelity, such as image generation or fine-grained reasoning. Another challenge is the *alignment between modalities*, where token granularity may need to be dynamically adjusted depending on the task—fine-grained tokens for generation and coarser tokens for understanding. The lack of *semantic alignment* across modalities can hinder multimodal integration, particularly when visual features do not correspond clearly to textual representations.

**Future Directions.** Future work should focus on *adaptive quantization mechanisms* that adjust token granularity based on the specific task or modality. Additionally, developing *unified token spaces* where both vision and language tokens can co-exist and interact seamlessly will be crucial for improving multimodal models. Incorporating *hybrid encoding schemes* that combine discrete tokens with continuous features will help maintain both high-level semantics and low-level details. Lastly, efforts should be directed towards improving *model controllability* and *explainability*, enabling more precise generation and better interpretability of the model's decisions.

## 4.2 Autoregressive-Driven Vision-Language Unifying

The narrative of autoregressive (AR) models in unified multimodal systems unfolds as a compelling extension of language modeling's triumphs into the visual realm. Originating from the sequential prediction paradigms that powered early successes in natural language processing—such as GPT's next-word forecasting—AR approaches adapted this causality to visuals, treating images and videos as token streams ripe for step-by-step construction. This evolution, as chronicled in recent surveys (Zhang et al., 2025c; Li et al., 2025a), marks a pivotal shift: from isolated visual tasks to a harmonious unification where understanding (e.g., scene reasoning) emerges as a byproduct of generative prediction. By modeling visuals autoregressively, these models not only generate coherent outputs but also infer semantics through the very act of forecasting, bridging the discrete worlds of pixels and words.

The story begins with foundational challenges: how to serialize high-dimensional visuals without losing fidelity, and how to fuse them with language for true multimodality. Early AR models tackled pixel-level generation, but as scales grew, innovations in tokenization and integration propelled unification forward. This section traces this arc through a structured classification, drawing on (Zhang et al., 2025c)'s emphasis on AR's role in large vision models (LVMs) and (Li et al., 2025a)'s insights into discrete generative paradigms. We categorize along three dimensions—representation strategies (encoding strategies), modality fusion mechanisms, and training methodologies—highlighting key milestones that transformed AR from a generative tool into a unified powerhouse.

It is useful to separate three levels that are sometimes conflated. **Visual tokenization** is the front-end operation that maps pixels or visual features into discrete indices or embeddings. **Unified token modeling** is the backbone-level decision to place visual and textual tokens in a common sequence and optimize them with the same or tightly coupled next-token objective. **Task formulation** specifies how a downstream task is expressed in that sequence. For example, VQA does not require the model to generate an image: the image is encoded into visual tokens, the question is appended as text tokens, and the answer is decoded as text. Image generation, by contrast, asks the same backbone to decode visual tokens that are later reconstructed by a visual decoder. This separation clarifies why a model may share an AR objective across tasks while still using different tokenizers, heads, or decoding formats for understanding and generation.

### 4.2.1 Fundamental Mechanism

Autoregressive visual generators model the joint distribution of a tokenized image $\mathbf{x} = (x_1, x_2, \ldots, x_N)$ as

$$p(\mathbf{x}) = \prod_{i=1}^{N} p(x_i \mid x_{<i}, c),$$

where $c$ denotes conditional inputs such as textual prompts, prior visual context, or multimodal embeddings. After generating a complete token sequence, a learned decoder (e.g., from a VQ-VAE or VQ-GAN) reconstructs the final image from discrete code indices. This causal formulation directly parallels the decoding process of large language models (LLMs), enabling a unified modeling interface for both textual and visual outputs. In recent unified models, AR modeling achieves unification by treating understanding and generation as variants of next-token prediction, naturally bridging discrete visual tokens with linguistic sequences.

### 4.2.2 Categories of Unified Autoregressive Models

Autoregressive (AR) unified models differ primarily in how they represent and serialize visual information before integration with language tokens. While Chapter 4 introduced the fundamental encoding paradigms—pixel-based, semantic-based, and hybrid tokenization—this section categorizes representative *unified AR models* according to the encoding strategy they adopt and how it defines their overall modeling behavior. Each category reflects a distinct trade-off among fidelity, alignment, and efficiency.

**(1) Pixel-Token Unified Models.** These models treat visual content as sequences of low-level discrete tokens obtained from pixel-oriented quantizers such as VQ-VAE or VQ-GAN. The tokens are serialized in raster order and processed autoregressively together with text tokens, forming a fully unified sequence-to-sequence pipeline. Representative works include Chameleon (Team, 2024), ANOLE (Chern et al., 2024), Emu3 (Wang et al., 2024c), UGen (Tang et al., 2025), and Liquid (Wu et al., 2024a). This design enables fine-grained visual reconstruction and photorealistic synthesis within a single Transformer backbone. However, pixel-token unification typically leads to extremely long token sequences—growing quadratically with image resolution—and the tokens themselves carry limited semantic abstraction. Consequently, while these models achieve high visual fidelity, they often struggle with cross-modal reasoning, long-context understanding, and computational scalability.

**(2) Semantic-Token Unified Models.** A second line of work encodes images using text-aligned encoders such as CLIP, SigLIP, or EVA-CLIP, transforming them into semantically meaningful discrete embeddings.

In this formulation, both visual and textual tokens inhabit a shared conceptual space, and the AR model operates directly on semantically aligned sequences. Representative systems include LaViT (Jin et al., 2023), DreamLLM (Dong et al., 2023), VL-GPT (Zhu et al., 2023b), and MetaMorph (Tong et al., 2025). This paradigm excels in cross-modal understanding and instruction following, as semantic embeddings align naturally with linguistic structure. However, the abstraction that enables strong reasoning also sacrifices spatial precision and pixel-level controllability, requiring separate diffusion or reconstruction decoders for high-quality generation. Thus, semantic-token unification emphasizes reasoning and understanding rather than direct synthesis.

**(3) Hybrid-Token Unified Models.** Hybrid models aim to combine the complementary strengths of pixel-level fidelity and semantic-level abstraction by fusing both types of tokens. Two primary variants exist: (a) *pseudo-hybrid* systems that employ distinct encoders for different tasks (e.g., semantic tokens for understanding, pixel tokens for generation), and (b) *joint-hybrid* systems that deeply integrate both token types into a single sequence space. Examples include Janus and Janus-Pro (Wu et al., 2025a; Chen et al., 2025b), OmniMamba (Zou et al., 2025), MUSE-VL, SemHiTok (Chen et al., 2025d), and Show-o2 (Xie et al., 2025). These approaches achieve more balanced performance between understanding and generation and demonstrate improved adaptability across diverse multimodal tasks. Nonetheless, they introduce new challenges in feature fusion, token redundancy, and training efficiency, as maintaining coherence across multiple token spaces remains non-trivial.

**(4) Summary and Discussion.** Autoregressive unified models have demonstrated the feasibility of handling visual and linguistic sequences within a single generative backbone. Yet, their architectural nature brings several inherent limitations:

- **Sequential inefficiency:** Token-by-token decoding causes inference latency to scale linearly with sequence length, which becomes prohibitive for high-resolution visual data.

- **Local generation without global refinement:** The strictly causal generation process prevents global feedback or holistic correction, leading to artifacts or incoherent layouts.

- **Semantic–structural imbalance:** Pixel-token models capture fine details but lack conceptual grounding, while semantic-token models achieve reasoning at the expense of controllable visual precision.

These issues motivate the exploration of *non-autoregressive paradigms*—particularly discrete diffusion models—that can perform **parallel, globally coherent, and semantically consistent** generation in a shared token space. The next section introduces this emerging direction, where discrete diffusion serves as a more flexible and efficient mechanism for unified vision–language modeling.

### 4.3 Discrete Diffusion-Driven Vision–Language Unifying

**Foundations.** In contrast to the autoregressive (AR) paradigm described in Section 4.2.2, diffusion-based modeling offers a fundamentally different view of generation. Instead of predicting the next token causally, diffusion reformulates generation as an *iterative denoising process*—a trajectory from structured noise to clean data. This paradigm is inherently non-sequential and global, enabling parallel decoding and holistic consistency.

Early works on continuous diffusion, such as DDPM and Latent Diffusion Models, have dominated pixel-level generation. However, extending diffusion to discrete token spaces—essential for unified vision–language modeling—required a series of conceptual breakthroughs. The seminal D3PM (Discrete Denoising Diffusion Probabilistic Model) (Austin et al., 2021) first introduced a categorical noise formulation:

$$q(x_t \mid x_{t-1}) = (1 - \beta_t)\,\mathrm{Cat}(x_{t-1}) + \beta_t\,p_{\mathrm{noise}}(x),$$

where $\beta_t$ controls the corruption rate and $p_{\mathrm{noise}}(x)$ defines the discrete noise prior (e.g., uniform or absorbing-state). This formulation preserves the Markovian structure of continuous diffusion while supporting discrete vocabularies such as text or visual tokens.

Subsequent work, SEDD (Score Entropy Discrete Diffusion) (Lou et al., 2023), refined this framework by introducing entropy-based training objectives for better stability and sample quality, establishing diffusion as a viable alternative to AR decoding in discrete domains. These advances bridged the conceptual gap between continuous image diffusion and token-based generative modeling.

Building upon these foundations, LLaDA (Large Language and Discrete Diffusion Alignment) (Nie et al., 2025) demonstrated that discrete diffusion can be scaled to LLM-level architectures. By treating both textual and visual tokens as elements of a shared categorical vocabulary, LLaDA unified multimodal denoising under a single Transformer backbone, revealing the potential of diffusion as a *non-causal, globally coherent unifier* for multimodal generation.

### 4.3.1 Fundamental Mechanism.

In unified discrete diffusion, the forward process defines a discrete Markov chain $\{x_t\}_{t=0}^T$ as above, and the reverse process—parameterized by a Transformer—predicts $p_\theta(x_{t-1} \mid x_t)$ at each step. Two corruption schemes are dominant:

- **Absorbing-state diffusion:** Tokens are replaced with a special absorbing symbol such as `[MASK]` with probability $\beta_t$. Once absorbed, a token remains masked until recovered during denoising. This design yields a well-defined termination state and smooth training dynamics.

- **Uniform-state diffusion:** Tokens are randomly replaced with uniformly sampled vocabulary items, producing a fully mixed state as $t \to T$. This variant simplifies the forward process but requires stronger modeling capacity in the reverse direction.

The model is trained via cross-entropy between predicted and ground-truth clean tokens, which serves as a discrete analogue of score matching.

**From Discrete Diffusion to Unified Multimodality.** The key insight enabling unified vision–language diffusion is that both visual and textual modalities can be represented as discrete tokens—e.g., from VQ-VAE or semantic tokenizers—thus sharing the same categorical domain. This allows a single diffusion Transformer to jointly denoise across modalities under *full attention*, enabling bidirectional information flow between text and vision. In contrast to AR models, diffusion updates all tokens in parallel at each timestep, facilitating:

- **Parallel decoding** — multi-token updates significantly reduce inference latency, especially for long visual sequences.

- **Full-context attention** — bidirectional attention allows global semantic reasoning and cross-modal coherence.

- **Iterative global refinement** — each denoising step refines prior outputs, ensuring consistent spatial and semantic alignment.

### 4.3.2 Evolution of Unified Diffusion Models.

Following the theoretical establishment of discrete diffusion, a new generation of models extended these ideas into large-scale unified vision–language systems: UniDisc (Swerdlow et al., 2025) pioneered the notion of unified discrete diffusion, introducing a masking-based denoising objective that jointly handles text and image tokens within one diffusion framework. It demonstrated controllable, non-autoregressive generation and editing, proving the feasibility of full unification. MMaDA (Yang et al., 2025b) advanced this paradigm toward reasoning-capable multimodal diffusion. By integrating chain-of-thought (CoT) reasoning and textual guidance during denoising, MMaDA bridged structured cognition and diffusion-based generation. Lumina-DiMOO (Xin et al., 2025) further scaled unified diffusion to the *"omni-modeling"* level, leveraging fully discrete tokenization and efficient sampling schemes to achieve high fidelity and sampling speed. Muddit (Shi et al., 2025) focused on improving visual realism by injecting priors from pre-trained text-to-image diffusion backbones, thus enhancing fidelity while maintaining unified control. Lavida-O (Li et al., 2025b) introduced

an Elastic-MoT (Mixture-of-Tasks) architecture that supports high-resolution (1024px) synthesis, object-level grounding, and compositional reasoning—signaling the maturation of unified discrete diffusion into scalable multimodal intelligence.

### 4.3.3 Summary and Outlook.

The progression from foundational discrete diffusion models (D3PM, SEDD) to large-scale unified frameworks (LLaDA, UniDisc, MMaDA, Lumina-DiMOO, Lavida-O) marks a shift from strictly sequential decoding toward globally refined multimodal generation. By reformulating autoregressive prediction as iterative denoising, diffusion-driven unification directly addresses specific AR bottlenecks—especially serial token latency and the lack of full-context refinement—when the tokenizer and denoising schedule are well trained. The current evidence is strongest at the conceptual and benchmark-demonstration level: these models enable *parallel inference*, *bidirectional semantic alignment*, and *iterative refinement*, but their downstream advantages still depend on tokenizer fidelity, sampling steps, and training stability. This paradigm provides practical flexibility for image synthesis, captioning, editing, and multimodal reasoning, while remaining less mature than established AR or continuous-diffusion recipes.

Despite these advances, the community surrounding discrete diffusion models is still evolving, and several challenges remain. Notably, the lack of a mature ecosystem for model training and experimentation means that getting robust results from scratch can be challenging. Training discrete diffusion models from zero is often less effective compared to pre-trained models, as the learning process can be highly sensitive to data quality, model architecture, and hyperparameters. Additionally, scaling models to handle more complex multimodal tasks can expose issues related to optimization stability and computational efficiency. These hurdles make it clear that while discrete diffusion is emerging as a powerful tool for holistic, controllable, and scalable vision-language intelligence, further research and community collaboration are needed to overcome the current limitations and refine these systems for broader use.

**AR vs. Discrete Diffusion (in prose).** AR decodes tokens sequentially with causal attention, yielding strong alignment with LLM training and simple control, but suffers from sequential latency and error accumulation on long visual sequences. Discrete diffusion updates tokens in parallel under full (bidirectional) attention, improving global coherence and speeding up long-sequence generation, but requires diffusion-specific objectives and currently has a less mature training ecosystem. We found this narrative clearer than a large table and it fits the page budget better.

### 4.4 Challenges of Discrete Unified Models

Discrete visual unified models, while aligning well with LLM architectures, face trade-offs in generative quality. The use of discrete representations often leads to a loss of fine visual details due to quantization, resulting in lower fidelity compared to continuous methods. This compromises the sharpness and realism of generated images.

Additionally, these models can struggle with handling complex visual details that require high-resolution generation, making them less effective in certain tasks where fine-grained outputs are crucial. Despite these drawbacks, the alignment with LLMs remains a key advantage, enabling more seamless multimodal integration and joint reasoning across vision and language tasks.

While these models offer a promising path towards scalable, unified frameworks, further research is needed to balance the trade-off between quality and multimodal alignment.

## 5 Datasets and benchmarks

### 5.1 Datasets for Unified Multimodal Models

A unified multimodal model's success crucially depends on the diversity and scale of its training data, which must support both perception and generation. These datasets can be broadly categorized into four major groups: (1) **Multimodal Understanding** datasets (e.g., LAION, COYO) for vision–language alignment;

(2) **Text-to-Image Generation** datasets (e.g., LAION-Aesthetics, Echo-4o-Image) for synthesizing high-quality visual content; (3) **Interleaved Image–Text** datasets (e.g., OBELICS, OmniCorpus) for providing document-level multimodal contexts; and (4) **Image Editing** datasets (e.g., InstructPix2Pix, AnyEdit) for controllable manipulation via instructions.

Table 2 summarizes representative datasets across these categories, which are often used in combination. Unified models typically rely on large-scale understanding corpora for grounding, generative datasets for synthesis, interleaved data for context reasoning, and editing datasets for controllable manipulation. The curation, filtering, and balancing of these multimodal datasets remain central to achieving robust unification.

Table 2: Representative Datasets for Unified Multimodal Models

| Category | Dataset Example | Ref. | Primary Contribution |
|---|---|---|---|
| **Understanding** | LAION / COYO | (Schuhmann et al., 2022b; Byeon et al., 2022) | Massive-scale (billions) web-scraped image-text pairs. |
| | DataComp | (Gadre et al., 2023) | Standardized large-scale (1.4B) data curation benchmark. |
| | LLaVA-OneVision | (Li et al., 2024b) | Large-scale (4.8M) instruction-following data. |
| **Generation** | LAION-Aesthetics | (Schuhmann et al., 2022b) | Filtered subset (120M) for high-quality T2I generation. |
| | AnyWord-3M / CosmicMan | (Tuo et al., 2023; Li et al., 2024d) | Focus on compositional reasoning and character consistency. |
| | Echo-4o-Image / BLIP-3o | (Ye et al., 2025; Chen et al., 2025a) | Instruction-tuned data for unified generative alignment. |
| **Interleaved** | OBELICS / OmniCorpus | (Laurençon et al., 2023; Li et al., 2024c) | Web-scale (141M / 8B) interleaved image-text documents. |
| | Multimodal C4 | (Zhu et al., 2023c) | Document-level multimodal contexts (101M documents). |
| **Editing** | InstructPix2Pix | (Brooks et al., 2023) | Paired (image, instruction, edited_image) data. |
| | SEED-Data-Edit / AnyEdit | (Ge et al., 2024; Yu et al., 2025) | Large-scale (3.7M+) instruction-driven edit corpora. |
| | ByteMorph | (Chang et al., 2025) | Fine-grained non-rigid deformation editing examples. |

Note: Results that involve closed-source systems or differing evaluation protocols may not be directly comparable; we report them for context and mark limitations in the text.

## 5.2 Benchmarks for Unified Multimodal Models

The evaluation of unified models requires benchmarks that span both understanding and generation. These benchmarks collectively form a continuous evaluation spectrum from perception to synthesis and are summarized in Table 3. Key evaluation categories include: (1) **Understanding**, which tests perception and reasoning (e.g., VQA, SEED-Bench); (2) **Image Generation**, assessing quality and controllability (e.g., DrawBench, T2I-CompBench); and (3) **Interleaved Generation**, which evaluates the ability to interleave text and visual reasoning (e.g., InterleavedBench, UniBench).

Table 3: Key Benchmarks for Unified Multimodal Models

| Category | Benchmark Example | Ref. | Primary Evaluation Focus |
|---|---|---|---|
| **Understanding** | VQA / OK-VQA | (Antol et al., 2015; Marino et al., 2019) | Visual question answering (standard and knowledge-based). |
| | MM-Vet / SEED-Bench | (Yu et al., 2023b; Li et al., 2023a) | Multi-domain, fine-grained multimodal comprehension. |
| | MathVista / General-Bench | (Lu et al., 2023a; Fei et al., 2025) | General-purpose reasoning (visual, textual, mathematical). |
| **Generation** | DrawBench / PartiPrompts | (Saharia et al., 2022; Yu et al., 2022b) | Prompt fidelity and complex scene composition. |
| | T2I-CompBench | (Huang et al., 2023) | Compositional reasoning (e.g., attributes, spatial relations). |
| | GenAI-Bench / WorldGenBench | (Li et al., 2024a; Zhang et al., 2025a) | Standardized protocols for T2I generation quality. |
| **Interleaved** | InterleavedBench | (Liu et al., 2024) | Human-curated samples of interleaved text and images. |
| | OpenLEAF / OpenING | (An et al., 2023; Zhou et al., 2025) | Open-domain query-response and mixed-modality generation. |
| | UniBench | (Li et al., 2025c) | Fine-grained evaluation of text-image-text compositions. |
| **Editing / Other** | MagicBrush | (Zhang et al., 2023b) | Instruction-based real-image editing. |
| | DreamBench++ | (Peng et al., 2024) | Subject-driven and identity-consistent generation. |
| | VTBench | (Lin et al., 2025b) | Visual tokenizer reconstruction quality (for discrete models). |

Note: Results that involve closed-source systems or differing evaluation protocols may not be directly comparable; we report them for context and mark limitations in the text.

## 5.3 Comparative Analysis of unified models on Evaluation Tasks

Before comparing benchmark scores, Tables 4 and 5 summarize the recent model landscape by representation and training objective. We include both strict unified understanding–generation models and engineering-

oriented unified generation systems, because the latter increasingly influence benchmark design and user expectations for image editing, subject consistency, and interleaved outputs.

**Extraction and comparability protocol.** Tables 6–8 are intended as a *survey map* rather than a single controlled leaderboard. We extracted numbers from the original papers, official technical reports, or released benchmark tables, and we retain each paper's own reporting convention unless a benchmark defines a standard aggregate score. We therefore compare rows only under the following caveats: (i) model scale, training data, instruction tuning, prompting templates, sampling parameters, and test-time tools may differ; (ii) some entries are closed-source or partially closed, preventing verification of data leakage, prompt rewriting, or evaluator versions; (iii) benchmark versions can drift across papers; and (iv) unified models sometimes report post-trained variants rather than base checkpoints. Consequently, we use these tables to identify *directional patterns*—for example, whether a representation family tends to favor captioning, compositional generation, or unified reasoning–generation—but we avoid treating small numerical gaps as statistically meaningful unless they come from the same paper or a controlled ablation. This protocol is particularly important for Tables 7 and 8, where prompt rewriting, self-CoT, model size, and proprietary evaluation pipelines can substantially change the reported score.

Table 4: Recent representative systems for unified multimodal understanding and generation. "Strict" denotes models that jointly support understanding and image generation within one tightly coupled framework.

| Model | Visual representation | Objective / scope | Key implication |
|---|---|---|---|
| Chameleon (Team, 2024) (2024) | Discrete image tokens | Next-token prediction; strict token unification | Establishes early-fusion mixed-modal modeling, but inherits visual-token quantization limits. |
| Emu3 (Wang et al., 2024c) (2024) | Discrete image/video tokens | Next-token prediction; strict token unification | Tests the strongest version of the "next-token only" hypothesis across understanding and generation. |
| Transfusion (Zhou et al., 2024) (2024) | Continuous VAE latent patches | LM loss + diffusion loss; strict hybrid objective | Shows continuous image patches and diffusion loss scale better than quantized image-token LM baselines. |
| Janus-Pro (Chen et al., 2025b) (2025) | Decoupled understanding/generation encoders | AR + visual generation; decoupled encoders | Demonstrates the practical value of separating visual paths while maintaining a unified interface. |
| BAGEL (Deng et al., 2025) (2025) | SigLIP2 features + FLUX VAE latents | Next-token + rectified flow; MoT route | Uses Mixture-of-Transformer experts and interleaved data to scale open unified modeling. |
| BLIP3-o (Chen et al., 2025a) (2025) | CLIP image features | Feature generation + visual decoding | Moves the generation target from VAE latents toward language-aligned semantic visual features. |
| UniFluid (Fan et al., 2025) (2025) | Continuous visual tokens + SigLIP prefix | Autoregressive continuous-token generation | Shows loss balancing and image-token order are central to continuous-token AR unification. |

Table 5: Additional recent systems emphasizing semantic encoders, pixel-space unification, and practical multimodal generation/editing. "Generation-centric" denotes systems that primarily unify visual generation/editing modes while often relying on separate understanding or conditioning modules.

| Model | Visual representation | Objective / scope | Key implication |
|---|---|---|---|
| UniWorld (Lin et al., 2025a) (2025) | High-resolution semantic encoder | Unified generation and editing | Argues that stronger semantic encoders can reduce the data burden for unified models. |
| OmniGen2 (Wu et al., 2025b) (2025) | Decoupled image tokenizer and pathways | Multimodal generation decoder; generation centric | Unifies text-to-image, editing, subject-driven, and in-context generation for practical deployment. |
| Mogao (Liao et al., 2025) (2025) | Dual vision encoders, interleaved context | Causal interleaved generation | Treats coherent text–image interleaved output as a primary capability. |
| Tuna-2 (Liu et al., 2026) (2026) | Raw pixel patch embeddings | Pixel-space flow matching + CE | Challenges the necessity of both VAE tokenizers and pretrained visual encoders. |

Note: Results that involve closed-source systems or differing evaluation protocols may not be directly comparable; we report them for context and mark limitations in the text. Table 6 presents the performance of unified multimodal models with continuous and discrete vision tokens across a diverse set of multimodal understanding benchmarks, including VQAv2 (Antol et al., 2015), OK-VQA (Marino et al., 2019), GQA (Hudson & Manning, 2019), NoCaps (Agrawal et al., 2019), Flickr30K (Young et al., 2014), MMMU (Yue et al., 2023), MMB (Liu et al., 2023b), and MME-P (Fu et al., 2023). Under the comparability protocol above, the safest conclusion is not that one representation family dominates universally, but that each family exposes a different bottleneck. Continuous-token models (e.g., BLIP-2, Emu, and Emu-I) are often strong on captioning-oriented tasks

Table 6: Comparison of unified multimodal models with continuous and discrete vision tokens for zero-shot understanding across multiple datasets. **Bold** is best and underline is second best.

| Model | VQAv2 | OK-VQA | GQA | NoCaps | Flickr | MMMU | MMB | MME-P |
|---|---|---|---|---|---|---|---|---|
| *continuous vision tokens* | | | | | | | | |
| BLIP-2 (Li et al., 2023c) | 65.0 | 45.9 | 44.7 | **121.6** | **97.6** | - | - | - |
| Emu (Sun et al., 2023b) | 52.0 | 38.2 | - | 96.5 | 72.0 | - | - | - |
| Emu-I (Sun et al., 2023b) | 57.2 | 43.4 | - | 108.8 | 77.4 | - | - | - |
| SEED-LLaMA (Li et al., 2023a) | 44.2 | 29.2 | - | - | - | - | - | - |
| SEED-LLaMA-I (Ge et al., 2024) | 66.2 | 45.9 | - | - | - | - | - | - |
| Nexus-Gen 7B (Zhang et al., 2025b) | **79.3** | - | - | - | - | 45.7 | - | 1602.3 |
| LLaVAFusion (Zhou et al., 2024) | - | - | - | - | - | 41.7 | - | 1603.7 |
| BAGEL (Deng et al., 2025) | - | - | - | - | - | 43.2 | 79.2 | 1610 |
| *discrete vision tokens* | | | | | | | | |
| Chameleon-MultiTask (Team, 2024) | 69.6 | - | - | - | 76.2 | - | - | - |
| Emu3 (Wang et al., 2024c) | 75.1 | - | 60.3 | - | 76.2 | 31.6 | 58.5 | 1243.8 |
| Liquid (Wu et al., 2024a) | 68.0 | - | 56.1 | - | - | - | - | 1119.3 |
| LaViT (Jin et al., 2023) | 66.0 | **54.6** | 46.8 | 114.2 | 83.0 | - | 58.0 | - |
| Dream-LLM (Dong et al., 2023) | 56.6 | 44.3 | 46.8 | 114.2 | 83.0 | - | - | - |
| VL-GPT (Zhu et al., 2023b) | 51.7 | 35.8 | 34.6 | - | - | - | - | - |
| Janus (Wu et al., 2025a) | 51.7 | 35.8 | 34.6 | - | - | 30.5 | 69.4 | 1338.0 |
| SemHiTok (Chen et al., 2025d) | - | - | 60.3 | - | - | 39.3 | 72.3 | 1449.0 |
| show-o (Xie et al., 2025) | 69.4 | - | 58.0 | - | 62.5 | 26.7 | - | 1097.2 |
| show-o2 (Xie et al., 2025) | - | - | **63.1** | - | - | **48.9** | **79.3** | **1620.5** |
| MMaDA (Yang et al., 2025b) | 76.7 | - | 61.3 | - | 67.6 | 30.2 | 68.5 | 1410.7 |

Note: Results that involve closed-source systems or differing evaluation protocols may not be directly comparable; we report them for context and mark limitations in the text.

Table 7: Evaluation of text-to-image generation ability on GenEval benchmark. **Bold** is the best and underline is the second best.

| Method | Single Obj. | Two Obj. | Counting | Colors | Position | Color Attri. | Overall↑ |
|---|---|---|---|---|---|---|---|
| *continuous vision tokens* | | | | | | | |
| SEED-X (Ge et al., 2024) | 0.97 | 0.58 | 0.26 | 0.80 | 0.19 | 0.14 | 0.49 |
| Nexus-Gen (Zhang et al., 2025b) | 0.99 | 0.86 | 0.53 | 0.85 | 0.78 | 0.59 | 0.77 |
| GPT-4o (Hurst et al., 2024) | 0.99 | 0.92 | **0.85** | **0.92** | 0.75 | 0.61 | 0.84 |
| BAGEL (Deng et al., 2025) | 0.99 | **0.94** | 0.81 | 0.88 | 0.64 | 0.63 | 0.82 |
| Skywork unipic (Wang et al., 2025b) | 0.98 | 0.92 | 0.74 | 0.91 | **0.89** | 0.72 | 0.86 |
| *discrete vision tokens* | | | | | | | |
| Chameleon (Team, 2024) | - | - | - | - | - | - | 0.39 |
| Emu3-Gen (Wang et al., 2024c) | 0.99 | 0.81 | 0.42 | 0.80 | 0.49 | 0.45 | 0.66 |
| Janus-Pro-7B (Chen et al., 2025b) | 0.99 | 0.89 | 0.59 | 0.90 | 0.79 | 0.66 | 0.80 |
| Janus (Wu et al., 2025a) | 0.97 | 0.68 | 0.30 | 0.84 | 0.46 | 0.42 | 0.61 |
| show-o (Xie et al., 2025) | 0.98 | 0.80 | 0.66 | 0.84 | 0.31 | 0.50 | 0.68 |
| show-o2 (Xie et al., 2025) | 0.99 | 0.86 | 0.55 | 0.86 | 0.46 | 0.63 | 0.73 |
| MMaDA (Yang et al., 2025b) | 0.99 | 0.76 | 0.61 | 0.84 | 0.20 | 0.37 | 0.63 |
| Lumina-DiMOO (Xin et al., 2025) | **1.0** | **0.94** | **0.85** | 0.89 | 0.85 | **0.76** | **0.88** |

where dense visual features and pretrained language decoders are beneficial. Discrete-token models (e.g., LaViT, Emu3, and show-o2) can be competitive on structured reasoning benchmarks when tokenized visual inputs are tightly integrated into the autoregressive backbone. However, these trends should be read as design signals rather than direct head-to-head rankings, because the entries differ in scale, training data, tuning recipes, and evaluation versions.

Table 7 further summarizes text-to-image generation results. Continuous-token models, including SEED-X, Nexus-Gen, GPT-4o, and BAGEL, often benefit from diffusion or flow-based decoders that preserve perceptual detail and support high-fidelity rendering. Discrete-token models such as Emu3-Gen, Janus-Pro-7B, show-o2, and Lumina-DiMOO show that categorical or mask-based visual modeling can also achieve

Table 8: Performance comparison on the unified understanding and generation benchmark WISE. **Bold** indicates the best result and underline indicates the second best.

| Model | Cultural | Time | Space | Biology | Physics | Chemistry | Overall |
|---|---|---|---|---|---|---|---|
| *continuous vision tokens* | | | | | | | |
| VILA-U-7b-256 (Lin et al., 2023) | 0.26 | 0.33 | 0.37 | 0.35 | 0.39 | 0.23 | 0.31 |
| MetaQuery-XL (Pan et al., 2025) | 0.56 | 0.55 | 0.62 | 0.49 | 0.63 | 0.41 | 0.55 |
| GPT-4o (Hurst et al., 2024) | **0.81** | **0.71** | **0.89** | **0.83** | **0.79** | **0.74** | **0.80** |
| BAGEL (Deng et al., 2025) | 0.44 | 0.55 | 0.68 | 0.44 | 0.60 | 0.39 | 0.52 |
| BAGEL w/ Self-CoT (Deng et al., 2025) | 0.76 | 0.69 | 0.75 | 0.65 | 0.75 | 0.58 | 0.70 |
| *discrete vision tokens* | | | | | | | |
| Janus-1.3B (Wu et al., 2025a) | 0.16 | 0.26 | 0.35 | 0.28 | 0.30 | 0.14 | 0.23 |
| JanusFlow-1.3B (Wu et al., 2025a) | 0.13 | 0.26 | 0.28 | 0.20 | 0.19 | 0.11 | 0.18 |
| Janus-Pro-1B (Wu et al., 2025a) | 0.20 | 0.28 | 0.45 | 0.24 | 0.32 | 0.16 | 0.26 |
| Janus-Pro-7B (Wu et al., 2025a) | 0.30 | 0.37 | 0.49 | 0.36 | 0.42 | 0.26 | 0.35 |
| Liquid (Wu et al., 2024a) | 0.38 | 0.42 | 0.53 | 0.36 | 0.47 | 0.30 | 0.41 |
| Emu3 (Wang et al., 2024c) | 0.34 | 0.45 | 0.48 | 0.41 | 0.45 | 0.27 | 0.39 |
| Orthus-7B-base (Kou et al., 2024) | 0.07 | 0.10 | 0.12 | 0.15 | 0.15 | 0.10 | 0.10 |
| Orthus-7B-instruct (Kou et al., 2024) | 0.23 | 0.31 | 0.38 | 0.28 | 0.31 | 0.20 | 0.27 |
| Show-o-demo (Xie et al., 2025) | 0.28 | 0.36 | 0.40 | 0.23 | 0.33 | 0.22 | 0.30 |

strong compositional scores. We qualify this observation because GenEval results may depend on prompt rewriting, sampling settings, checkpoint scale, and whether the reported system is a base model or a post-trained variant.

The trend becomes more nuanced in unified evaluation. As shown in Table 8, models with continuous or hybrid continuous pathways perform strongly on WISE, especially when spatial, temporal, and physics-related reasoning must be coupled with generation. This does not prove that continuous representations are intrinsically superior in all unified settings; rather, it suggests that current continuous or hybrid systems better preserve visual detail and support self-consistency mechanisms needed by WISE-like tasks. Discrete-token systems still offer advantages in language-model compatibility and controllable token-level operations, but may require stronger tokenizers, refinement mechanisms, or hybrid semantic features to reduce information bottlenecks.

Taken together, these results reveal a fundamental trade-off across representation paradigms. Discrete vision tokens provide direct compatibility with language modeling and token-level control, while continuous tokens preserve richer visual information and naturally support diffusion/flow decoders. The empirical evidence is heterogeneous, so our main claim is a design-level synthesis: discrete methods should prioritize tokenizer fidelity and global refinement, whereas continuous methods should prioritize semantic alignment, attention cost control, and feedback between the reasoning and generation pathways.

This observation highlights a key challenge for future unified multimodal models: designing representation mechanisms that combine the compositional advantages of discrete tokens with the semantic richness of continuous features, thereby enabling both strong reasoning and high-fidelity generation within a single unified framework.

## 5.4 Challenges and Future Directions in Benchmarking

Despite the growing diversity of benchmarks, the evaluation of unified multimodal models remains fragmented. Existing benchmarks are often modality-imbalanced (neglecting audio or video), and automated metrics (e.g., FID, CLIP) fail to capture compositional correctness or factual grounding. Most benchmarks still adopt static, single-turn tasks, whereas real-world unified agents require evaluation in dynamic, interactive, and long-horizon multimodal contexts. Future benchmarking should move toward open, adaptive frameworks that integrate human–AI co-evaluation, contextual reasoning, and unified scoring across perception, reasoning, and generation. Such comprehensive benchmarks will be key to tracking genuine progress toward general-purpose multimodal intelligence.

# 6 Challenges and opportunities

The development of unified multimodal models that integrate both understanding and generation remains in its infancy. While recent advances have demonstrated the feasibility of bridging autoregressive (AR) and diffusion paradigms, significant challenges persist across architectural, algorithmic, and data dimensions. In particular, it remains unclear whether multimodal understanding and generation constitute competing objectives that impose unavoidable trade-offs, or whether they can be jointly optimized in a synergistic manner as model capacity and data scale increase. At the same time, these challenges open new opportunities for innovation and cross-domain applications.

## 6.1 Whether Understanding and Generation Tasks Improve Each Other?

A central question in unified multimodal modeling is whether visual understanding and visual generation inherently compete for model capacity, or whether they can instead reinforce each other under appropriate design choices. Early multimodal systems often treated understanding and generation as loosely coupled or sequential processes, assuming that joint training would lead to negative interference. However, recent fully unified models (Tong et al., 2025; Wu et al., 2024a; An et al., 2025; Zhu et al., 2023b) provide increasing evidence that understanding and generation are not independent capabilities, but can mutually reinforce each other when jointly optimized.

Liquid (Wu et al., 2024a) reveals that a trade-off between language generation and visual generation does exist at smaller model scales, where limited representational capacity constrains simultaneous optimization. Notably, this trade-off is not fixed. As model size increases, the performance gap between unimodal and unified training progressively narrows, and in some cases nearly vanishes.

Liquid (Wu et al., 2024a) further conducts controlled experiments, showing that adding understanding data consistently improves generation performance, while incorporating generation data also enhances understanding metrics under the same baseline. The paper attributes this phenomenon to the alignment of optimization objectives within a unified multimodal space. UniCTokens (An et al., 2025) confirms this trend by systematically comparing unified models against understanding-only and generation-only counterparts, showing that unified training yields superior performance, particularly on tasks requiring semantic understanding and generation consistency.

Similarly, MetaMorph (Tong et al., 2025) reports strong bidirectional gains: fixing the amount of generation data and increasing VQA data improves both understanding and generation, and vice versa. Notably, joint training significantly improves sample efficiency for generation, enabling models to acquire stable visual token generation with only a few thousand generation samples.

These results suggest that understanding and generation share substantial underlying structures, including vision–language alignment, semantic consistency constraints, and visual token formation mechanisms. Understanding tasks provide structured semantic supervision that stabilizes and constrains generation, while generation tasks force models to explicitly model fine-grained visual attributes and compositional details that are often underrepresented in understanding-only training. When optimized within a shared representation space, these two processes form a positive feedback loop, alleviating information sparsity and representation bias commonly observed in single-task training.

Significantly, MetaMorph further analyzes whether certain visual understanding tasks correlate more strongly with generation performance. It reports that tasks emphasizing general visual comprehension and vision-centric grounding exhibit the strongest correlation with visual generation quality, whereas knowledge-heavy understanding tasks contribute comparatively weaker gains. This finding indicates that visual generation benefits primarily from understanding signals that reinforce perceptual alignment and grounded visual representations, rather than relying on abstract or symbolic reasoning alone.

Overall, these findings challenge the conventional view that understanding and generation are competing objectives. Instead, they support a unifying perspective in which understanding and generation become complementary processes under sufficient model capacity and aligned modality representations. From this

standpoint, their mutual reinforcement is not an incidental artifact of training, but a natural consequence of unified multimodal modeling.

## 6.2 Challenges

Building upon the advances reviewed in the previous sections, this part focuses on the major challenges that remain unresolved in developing unified multimodal understanding and generation models. These challenges reveal the current limitations of existing methods and highlight areas where further innovation is required.

**1) Visual Tokenization and Representation.** A core difficulty lies in effectively representing visual information in a tokenized form suitable for autoregressive generation. Discrete tokenization methods based on VQ-VAE or VQ-GAN enable alignment with textual tokens but often sacrifice fine-grained visual details. Conversely, continuous representations preserve semantic richness but complicate sequence modeling and decoding. Achieving a unified tokenization scheme that balances efficiency, fidelity, and semantic expressiveness remains an open problem.

**2) Architectural Divergence Between Understanding and Generation.** Multimodal understanding models are primarily autoregressive, focusing on sequential reasoning and text prediction, whereas visual generation models rely heavily on diffusion processes for iterative denoising and synthesis. These fundamentally different modeling paradigms challenge architectural unification. Designing hybrid models that can seamlessly switch or jointly optimize both reasoning and generation remains a key research frontier.

**3) Cross-Modal Alignment and Attention Mechanisms.** Unified models must integrate heterogeneous modalities—text, image, video, and possibly audio—within a shared latent space. Scaling cross-modal attention to high-resolution visual inputs introduces substantial computational and memory overhead. Furthermore, maintaining semantic coherence across modalities, especially in complex tasks like instruction-based image generation or multimodal reasoning, poses an additional challenge.

**4) Data Scarcity and Benchmark Limitations.** Constructing large-scale, high-quality datasets that jointly support understanding and generation tasks is still difficult. Existing datasets are either biased toward single-modal tasks (e.g., captioning or VQA) or limited to generative tasks (e.g., text-to-image). Moreover, there is a lack of standardized benchmarks that simultaneously evaluate reasoning accuracy, visual quality, and cross-modal consistency. Comprehensive evaluation frameworks are urgently needed.

**5) Computational Cost and Scalability.** Unified multimodal models require vast computational resources due to their multimodal encoders, diffusion decoders, and autoregressive backbones. Training such large-scale systems demands extensive data and compute budgets, which limits accessibility and reproducibility. Efficient architectures, model compression, and modular training pipelines are essential for scalable deployment.

## 6.3 Opportunities

Despite these challenges, the pursuit of unification offers remarkable opportunities for advancing multimodal AI research and applications.

**1) Toward General-Purpose Multimodal Intelligence.** A successful unified model can both *understand* and *generate* multimodal content, enabling end-to-end perception and creation within a single framework. Such models could serve as general-purpose agents capable of reasoning over complex visual scenes and producing coherent, high-quality visual outputs in response.

**2) Emergent Capabilities Through Modality Fusion.** Unifying understanding and generation may yield emergent capabilities that neither paradigm can achieve alone. For instance, reasoning-augmented image generation can improve semantic controllability, while generation-enhanced understanding can enable visual imagination and explanation in natural language reasoning tasks.

**3) Architectural Innovation.** The coexistence of autoregressive and diffusion mechanisms encourages the exploration of new hybrid frameworks that leverage the strengths of both paradigms. Promising directions

include bidirectional AR-diffusion coupling, shared latent token spaces, and unified decoders capable of multimodal synthesis.

**4) Advancement of Datasets and Evaluation Protocols.** The unification trend motivates the construction of novel datasets encompassing both understanding and generation signals, as well as comprehensive benchmarks evaluating semantic accuracy, visual quality, and reasoning consistency. This will foster more standardized and comparable research progress.

**5) Practical Applications and Ecosystem Growth.** Unified models are poised to power next-generation applications in interactive AI, visual storytelling, robotics, design, and education. Their ability to reason and generate across modalities will enable seamless multimodal interaction and content creation, expanding the reach of AI systems into creative and scientific domains.

## 6.4 Cross-Axis Meta-Analysis

We synthesize trends across *token type* (continuous vs. discrete) and *interaction* (serial vs. parallel) in terms of concrete design consequences. Continuous tokens are strongest when fidelity, editing smoothness, and diffusion priors dominate the target application; their failure mode is that semantic alignment must be learned through adapters, shared blocks, or auxiliary objectives. Discrete tokens are strongest when one wants a single sequence-modeling interface and token-level control; their failure mode is that quantization error and long visual sequences can harm fine detail and latency. Serial coupling is attractive for deployment because each module can be upgraded independently, but its static conditioning creates a semantic–visual gap. Parallel coupling reduces this gap by allowing language and visual states to interact during generation, but it pays for this with longer joint contexts, more complex attention masks, and harder optimization.

This cross-axis view yields four concrete takeaways. First, if the main bottleneck is *photorealism or local edit fidelity*, continuous latent diffusion remains the safest default. Second, if the main bottleneck is *instruction following, symbolic control, or interleaved text–image output*, discrete or hybrid tokenization is often preferable. Third, if the main bottleneck is *semantic–visual mismatch*, parallel or feedback-enabled coupling is more important than the choice of tokenizer alone. Fourth, if the main bottleneck is *latency or memory*, the decisive variables are token count, attention pattern, and denoising/decoding steps rather than the continuous/discrete label itself.

Evidence remains heterogeneous across closed/open systems; we therefore report qualitative trends and call for standardized, open unified benchmarks with shared evaluation protocols and ablations (e.g., token length, attention schedule, coupling strength).

## 6.5 Roadmap and Concrete Methods

Below we outline concrete, ready-to-implement avenues with brief recipes and pitfalls:

- **Hybrid/Hierarchical tokenization.** *Recipe:* pretrain a VAE (continuous latents) for fidelity, then stack a VQ codebook over mid-level latents for discrete alignment (e.g., FSQ/VQ improvements (Mentzer et al., 2023; Zheng et al., 2022; Zhu et al., 2024)); expose both streams to the LLM via gated adapters. *Benefits:* short continuous sequences for generation, discrete hooks for reasoning/control. *Pitfalls:* codebook collapse; mitigate via utilization regularizers and temperature annealing.

- **Shared latent spaces for understanding–generation.** *Recipe:* tie the vision encoder used for understanding with the generator's latent space via contrastive/alignment losses and shared blocks; allow gradients from generation to update reasoning states during curriculum phases (Li et al., 2023c; Sun et al., 2023b; Ge et al., 2024). *Measure:* alignment via retrieval, caption faithfulness, and edit consistency.

- **AR–Diffusion coupling schedules.** *Recipe:* alternate K diffusion denoising steps with one AR language step (or share cross-attention blocks) so that evolving text states inform denoising and vice versa (Zhou et al., 2024; Zhao et al., 2024; Shi et al., 2024). *Stability:* warm-start with frozen text/vision backbones, then unfreeze in stages; use low-rank adapters on shared blocks.

- **Parallel decoding.** *Recipe:* for AR, use multi-token prediction and speculative decoding; for discrete diffusion, mask-and-refine with schedule-aware guidance (Pang et al., 2024; Austin et al., 2021). *Trade-offs:* larger context windows improve consistency but raise memory; prefer blockwise attention.

- **Unified curricula.** *Recipe:* stage data from captioning/image QA $\rightarrow$ compositional/constrained synthesis $\rightarrow$ open-ended interleaved tasks; interleave editing data to tighten controllability. Track per-stage win/loss on a shared held-out suite (e.g., VQA+GenEval+WISE) with ablations on token length and attention schedule.

- **Evaluation and reporting.** *Minimum set:* (i) understanding: VQA-style + compositional reasoning; (ii) generation: text–image faithfulness and attribute tests; (iii) unified: interleaved reasoning+generation (e.g., WISE-like). Report latency (prompt $\rightarrow$ image), memory, and cost per 1k tokens/steps for reproducibility.

## 6.6 Computational Cost Trade-offs

We summarize qualitative and approximate cost factors observed in practice. Let $N_t$ denote text length, $N_v$ visual token length, $d$ hidden size, $L$ transformer layers, and $K$ diffusion or refinement steps. A joint full-attention block has approximate attention cost $O(L(N_t + N_v)^2 d)$, while a causal AR visual decoder has decoding latency that scales with $N_v$ sequential steps even when each step uses cached attention. Diffusion or discrete diffusion reduces sequential dependence by updating many visual tokens in parallel, but pays $K$ iterative denoising/refinement passes. Serial coupling can keep the LLM context short and offload generation to a specialized visual model; parallel coupling improves feedback but expands the joint context. Thus, the computational trade-off is governed by *visual token length, attention span, number of denoising/refinement steps,* and *whether the visual tokenizer/decoder is frozen or jointly trained.*

Table 9: Computational trade-offs across tokenization and interaction paradigms. $N_v$ is visual token length and $K$ is the number of diffusion/refinement steps. Entries are approximate and intended for comparing scaling tendencies rather than exact wall-clock cost.

| Paradigm | Dominant cost | Latency scaling | Memory scaling | Main design implication |
|---|---|---|---|---|
| Continuous + Serial | Frozen/fine-tuned VAE or diffusion decoder; LLM-to-generator connector | $O(K)$ denoising after a compact semantic plan | LLM context can be short; visual memory mostly in generator | Efficient to deploy and upgrade, but weak feedback from generated pixels to language reasoning. |
| Continuous + Parallel | Joint language–latent attention plus diffusion/flow prediction | $O(K)$ denoising with larger joint context | $O((N_t + N_v)^2)$ attention if full joint attention is used | Better semantic–visual feedback, but expensive at high resolution unless using sparse/block attention. |
| Discrete + AR | Long visual token sequence and token-by-token decoding | $O(N_v)$ sequential visual decoding; cached attention reduces but does not remove serial latency | $O((N_t + N_v)^2)$ training attention; cache grows with sequence length | Mature LLM recipe and token-level control, but slow for high-resolution visual outputs. |
| Discrete Diffusion | Iterative mask/noise refinement over discrete tokens | $O(K)$ parallel refinement, often with $K \ll N_v$ | $O((N_t + N_v)^2)$ per refinement pass unless sparsified | Reduces AR latency and enables global correction, but training recipes and evaluator conventions are less mature. |
| Hybrid/MoT | Multiple token streams or task-specific experts | Depends on routed path; can activate only a subset of experts | Router/expert memory plus active-token attention | Useful compromise: semantic tokens for reasoning, generative latents for fidelity, at the cost of routing complexity. |

**Reporting protocol.** We recommend reporting: (i) end-to-end latency (prompt $\rightarrow$ image/video) with resolution, batch size, and hardware; (ii) peak memory; (iii) $N_t$, $N_v$, attention pattern, and $K$; (iv) tok-

enizer/decoder cost and whether these modules are frozen; (v) training tokens, GPU-hours, and post-training data; (vi) unit cost (per 1k tokens, per image, or per video second); and (vii) failure modes (hallucination, attribute leakage, edit drift, temporal inconsistency) on a shared suite (e.g., VQA (Antol et al., 2015), T2I-CompBench (Huang et al., 2023), InterleavedBench (Liu et al., 2024)). This protocol makes computational claims falsifiable and prevents small benchmark gains from obscuring large differences in scale or test-time compute.

## 6.7 Industrial Systems and Applications

This subsection summarizes practical deployment patterns and caveats for unified VLMs in production; citations indicate representative open benchmarks rather than endorsements of closed systems.

**What production optimizes for.** Five axes: (1) *Task mix* (drafting vs. final render vs. editing; e.g., MagicBrush/DreamBench++ (Zhang et al., 2023b; Peng et al., 2024), OpenLEAF/InterleavedBench (An et al., 2023; Liu et al., 2024)); (2) *Latency* (UI draft <1s; refined 2–10s); (3) *Safety* (guardrails, approvals); (4) *Cost* (steps/tokens/cache); (5) *Traceability* (provenance, logs).

**Recommended configurations by use case.**

- **Creative co-pilots:** Draft with Discrete+AR for fast edits; refine with Continuous+Diffusion for fidelity and global consistency.

- **Document/slide agents:** Discrete+AR or Discrete Diffusion; optional continuous latents for complex figures; emphasizes layout reasoning and parallel refinement.

- **Education/accessibility:** Discrete+AR with strict guardrails; optional continuous for diagrams; prioritizes refusals and compositional explanations.

- **Enterprise brand workflows:** Discrete+AR with style constraints; Continuous for approval renders; ensures traceability and high-quality deliverables.

**Practical playbook.** (1) *Draft*: produce an immediate AR-based preview; cache encoder features to respect latency. (2) *Refine*: switch to diffusion (continuous or discrete) or mask-and-refine for global consistency. (3) *Safety*: input/output filters, red-team suites, refusal consistency checks. (4) *Delivery*: provenance/watermark, audit logs, acceptance testing.

**Reporting.** Follow the protocol in Sec. 6.6 (latency, memory, token/step counts, unit cost, failure modes) and separate closed-source metrics from open benchmarks.

## 6.8 Summary

In summary, while unified multimodal understanding and generation models face nontrivial challenges in representation, architecture, data, and scalability, they also present transformative opportunities for the future of AI. Continued exploration of hybrid architectures, efficient training strategies, and standardized evaluation will accelerate progress toward truly general multimodal intelligence.

# 7 Summary and Future Directions

**Synthesis.** Unified vision–language modeling benefits from tight coupling between understanding and generation. Our taxonomy (continuous vs. discrete visual tokens; serial vs. parallel interaction) reveals complementary strengths: continuous latents favor fidelity and holistic reasoning; discrete tokens align with LLM training and controllability; parallel coupling improves bidirectional coherence at higher compute.

**What we know.** Evidence across recent systems suggests that (i) understanding and generation can be mutually reinforcing when optimized in a shared space; (ii) hybrid or hierarchical tokenization helps reconcile fidelity with control; (iii) AR–diffusion coupling and masking-based refinement reduce sequential

bottlenecks; (iv) evaluation must span perception, composition, and interleaved reasoning, with reporting of latency, memory, and token/step counts.

**Open problems.** (1) Unified tokenization: balance efficiency, fidelity, and semantic expressiveness without codebook collapse or information loss. (2) Training stability at scale: schedule alignment between reasoning updates and generative refinement; robust curricula across tasks. (3) Cost and latency: reduce long-sequence overheads (multi-token prediction, blockwise attention, cache reuse). (4) Safety and traceability: defense-in-depth for multimodal prompts, provenance and watermarking in production settings.

**Actionable roadmap.** Hybrid tokenizers (continuous base + discrete hooks), shared latent spaces with gradient feedback from generation, coupled AR–diffusion schedules (alternating steps or shared blocks), and parallel decoding (multi-token AR; mask-and-refine for discrete diffusion). We recommend a unified reporting protocol (latency/memory/tokens/steps/unit cost/failure modes) and adoption of interleaved benchmarks (e.g., InterleavedBench, OpenLEAF) alongside VQA and T2I compositional suites.

**Looking ahead.** We anticipate a gradual convergence toward omni-capable backbones featuring (i) mixed discrete–continuous representations, (ii) bidirectional fusion throughout decoding, and (iii) evaluation that emphasizes reliability under distribution shift and instruction-following faithfulness. Industrial deployments should prefer two-stage draft–refine workflows with explicit safety gates and audit trails.

## 8 Ethics and Safety

With the rapid advancement of Unified Vision-Language Models (Unified VLMs), integrating visual understanding and generation within a single architecture has emerged as a central paradigm in multimodal artificial intelligence (Li et al., 2023c). By jointly modeling visual and linguistic information in a shared representation space (Linear-probe, 2021), these models enable a tight coupling between perception and generation, where understanding informs generation and generative processes, in turn, enhance interpretability (Zhou et al., 2019) (Yu et al., 2022a). This unified framework has demonstrated strong capabilities in complex instruction following (Liu et al., 2023a) (Wang et al., 2024a), interleaved text–image generation, and long-horizon multimodal reasoning. However, while such cross-modal unification substantially improves model capacity and flexibility, it also introduces more intricate and systemic challenges in terms of ethics and safety (Weng et al., 2025) (Chen et al., 2025c) (D'Antonoli et al., 2025).

### 8.1 Data Bias: Inheritance, Amplification, and Measurement

Unified VLMs are typically trained on large-scale image–text corpora, (Schuhmann et al., 2022a) (Gadre et al., 2023). Despite their massive scale, these datasets often exhibit substantial distributional biases, including the over-representation of Western cultures, dominant social groups, and specific demographic attributes, while underrepresenting or mischaracterizing marginalized populations (Kay et al., 2015) (Kärkkäinen & Joo, 2021) (Devries et al., 2019).

Within a unified modeling framework, such biases are not only inherited but can also be amplified through cross-modal interactions (Bender et al., 2021) (Liu et al., 2025). On the one hand, in understanding tasks, models may exhibit systematic misclassification or negative labeling tendencies toward certain groups (Devries et al., 2019) (Xiao et al., 2024b). On the other hand, in generative tasks, these biases manifest as stereotypical associations between social roles and demographic attributes—for instance, generating male figures for occupations such as "doctor" or "manager," while associating roles like "waiter" with female or minority groups (Bolukbasi et al., 2016) (Wang et al., 2024b).

Existing studies have shown that increasing model scale does not necessarily mitigate bias; instead, it may exacerbate such issues (Kaplan et al., 2020) (Kärkkäinen & Joo, 2021). In large-scale vision-language models, certain demographic groups have been found to be disproportionately associated with negative concepts (e.g., criminality) (Kärkkäinen & Joo, 2021) (Gwilliam et al., 2021), while the model's ability to distinguish minority groups degrades (Devries et al., 2019), sometimes leading to phenomena akin to category collapse. Moreover, although earlier models exhibited explicit errors in human versus non-human classification, such

issues have increasingly shifted toward more subtle forms of bias, such as implicit associations with negative social attributes (May et al., 2019) (Nadeem et al., 2020) (Nangia et al., 2020).

To address these challenges, a range of bias measurement and mitigation strategies have been proposed. For example, the Multimodal Composite Association Score (MCAS) quantifies bias by measuring the strength of associations between visual and textual embeddings (Mandal et al., 2023). Mitigation approaches include dataset rebalancing and filtering, inference-time calibration, and counterfactual training, all aiming to reduce spurious correlations during both training and generation (Kim et al., 2018) (Chuang et al., 2023).

## 8.2 Environmental Impact: Energy, Carbon, and Resource Consumption

The training and deployment of Unified VLMs typically require substantial computational resources, making their energy consumption and environmental impact an increasingly important concern (Menghani, 2021). Owing to the integration of large-scale vision encoders and billion- to trillion-parameter language models, the associated training process incurs significant electricity usage and carbon emissions (Strubell et al., 2019).

Recent studies estimate that training a state-of-the-art model may consume energy comparable to the annual electricity usage of hundreds of households (Patterson et al., 2022), with a non-negligible carbon footprint. In addition, large-scale data centers introduce further environmental costs through cooling demands, leading to increased water consumption and amplifying the overall lifecycle impact of these systems (Li et al., 2023e).

To mitigate these environmental concerns, current research has explored several directions. First, Mixture-of-Experts (MoE) architectures reduce computational overhead by activating only a subset of parameters during each forward pass (Fedus et al., 2021). Second, smaller-scale multimodal models leverage techniques such as knowledge distillation to retain competitive performance at significantly lower parameter counts (Hinton et al., 2015) (Jiao et al., 2019). Third, carbon-aware scheduling strategies dynamically adjust training workloads based on the carbon intensity of the energy supply (Radovanovic et al., 2021). Finally, advances in hardware efficiency further improve performance per watt (Jahns et al., 2025) (Jouppi et al., 2023).

Despite these efforts, the continuous scaling of model size and training data tends to offset a substantial portion of these gains, leaving the overall environmental footprint an open challenge.

## 8.3 Harmful Content Generation and Multimodal Safety Alignment

While the cross-modal capabilities of Unified VLMs enhance interaction flexibility, they also introduce new security risks (Li et al., 2023b). Compared to traditional text-only models, these systems can exploit visual inputs to bypass existing safety mechanisms (Jiang et al., 2025) (Luo et al., 2024), thereby generating inappropriate or harmful content.

Representative attack vectors include typographic attacks, where malicious instructions are embedded within images to evade text-based filtering; multimodal prompt attacks that implicitly convey harmful intent through visual compositions (Waseda et al., 2025) (Lee et al., 2025); and adversarial perturbations that manipulate the model's perception pathway via imperceptible input modifications (Zhang et al., 2025d). These attacks exploit inherent vulnerabilities in cross-modal fusion, significantly increasing the success rate of jailbreak attempts (Weng et al., 2025).

To address these challenges, recent work has explored safety alignment techniques tailored for multimodal settings (Bai et al., 2022). For instance, Safe RLHF-V introduces dual-objective optimization during reinforcement learning by jointly modeling helpfulness and safety constraints, enabling a more balanced trade-off between utility and risk (Dai et al., 2023) (Perez et al., 2022). In addition, multi-level guardrail mechanisms perform hierarchical filtering and re-ranking of model outputs to reduce the likelihood of harmful generations (Rebedea et al., 2023). Another line of work transforms visual inputs into structured textual representations, allowing unified safety auditing pipelines and improving overall system robustness (Li et al., 2023d).

## 8.4 Representation-Specific Risk Profile

The representation-centric taxonomy used in this survey also changes the safety analysis. **Continuous-token systems** inherit the strengths and risks of diffusion or flow decoders: they can generate high-fidelity and editable images, which increases misuse risk for photorealistic misinformation, identity manipulation, and subtle visual edits. Their latent spaces are often less interpretable, making provenance, watermarking, and latent-level auditing important. **Discrete-token AR systems** expose visual content as language-like token sequences; this improves controllability and logging, but also creates new prompt-injection and jailbreak surfaces because malicious visual or textual tokens can be interleaved in the same context. **Discrete diffusion systems** add mask-and-refine or parallel denoising steps, so safety filters should audit intermediate refinement states rather than only the final output. **Parallel-coupled systems** are especially powerful but risky because understanding and generation can reinforce each other: a successful malicious instruction can be interpreted, planned, and rendered inside one feedback loop. These distinctions suggest that safety evaluation should be reported by representation and coupling type, not only by the final application.

## 8.5 Mitigation and Evaluation Checklist

For unified models, we recommend the following minimum checklist with concrete tests:

- **Misinformation/Deepfakes:** watermarking/provenance (self-report + third-party verification); prompt suites for person identity swaps, staged events, and compositional claims. Measure detector AUC and false refusal.

- **Data Bias:** dataset cards + demographic coverage; counterfactual augmentation; MCAS and group-wise accuracy gaps with acceptance thresholds and confidence intervals.

- **Environmental Impact:** report FLOPs, wall-clock, energy and carbon (with source mix); compare to MoE/distillation baselines at matched quality.

- **Safety/Jailbreaks:** defense-in-depth (input filters, safety heads, output re-ranking); red-team suites for typographic/visual prompt attacks and gradient-based perturbations; log refusal consistency.

- **Unified Eval:** end-to-end tasks combining reasoning+generation (e.g., WISE-like), plus editing controllability and instruction-following with factuality checks; release scripts and seeds.

## 8.6 Ethical Summary and Outlook

Overall, while Unified VLMs have significantly advanced multimodal intelligence, they also transform ethical and safety concerns from isolated unimodal risks into cross-modal, system-level challenges (Bommasani et al., 2021). Among these, data bias, environmental cost, and multimodal safety vulnerabilities have emerged as central issues in current research (D'Antonoli et al., 2025).

Looking forward, future work should strike a better balance between model performance and social responsibility. On the one hand, it is crucial to develop data curation and training paradigms that are more fair, interpretable, and controllable (Barocas et al., 2018). On the other hand, there is a growing need for unified safety evaluation and defense frameworks that generalize across multimodal settings (Perez et al., 2022). Meanwhile, green computing practices and efficient model design will play a key role in enabling the sustainable development of unified multimodal systems (Patterson et al., 2022).

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
