# OpenReview forum: "Unifying Understanding and Generation in Vision-Language Models: Advances, Challenges, and Opportunities"
_TMLR — Decision pending for TMLR_

### Review · Reviewer_feLg · 2026-03-16

**Summary Of Contributions:**

This paper provides a systematic survey of unified VLMs that integrate discriminative understanding and generative capabilities. The authors propose a taxonomy based on visual representation: Continuous Tokens (subdivided into Serial and Parallel Coupling) and Discrete Tokens (comparing Autoregressive vs. Discrete Diffusion). The work effectively maps the evolution of the field from 2021 to 2025, providing a comprehensive reference for architectures, datasets, and evaluation benchmarks.

Strengths:

- Highly timely, covering state-of-the-art models (e.g., Janus, Transfusion).
- Clear and logical taxonomy centered on the visual-semantic gap.

Weaknesses:

- Lack of unified quantitative comparisons across models.
- Lack of ethical and broader impact discussions.
- Lack of discussion on industrial solutions and applications.

**Audience:**

Yes

**Audience Explanation:**

Most of the audience working on visual understanding and generation would be interested in reading this paper.

**Broader Impact Concerns:**

The current submission entirely lacks a discussion on the ethical implications of unified VLMs. This is a significant omission for TMLR. The authors should at least address the following points:

- Misinformation and Deepfakes: How the seamless integration of understanding and generation could be misused for more sophisticated AI-generated misinformation.
- Data Bias: The risk of unified models inheriting and amplifying societal biases present in large-scale web-crawled datasets.
- Environmental Impact: The carbon footprint associated with training and scaling these massive unified architectures.
- Safety/Harmful Content: Mechanisms for preventing the generation of toxic content or responding to malicious multimodal prompts.

**Claims And Evidence:**

Yes

**Claims Explanation:**

The categorization and technical descriptions are well-supported by an extensive bibliography of recent SOTA works. The paper accurately reflects the current shift in the community toward native multimodal unification.

**Requested Changes:**

1. Add a section addressing the ethical challenges specific to unified models (see Broader Impact Concerns).
2. Briefly discuss the trade-offs in computational cost between continuous and discrete tokenization strategies.
3. Authors messed up with Citep and Citet. All the citation formats must be updated.
4. Add an analysis on industrial solutions and applications.

---

> ### Author Response · Authors · 2026-04-06
> **Review for Reviewer feLg**
>
> We appreciate your constructive feedback. We have implemented concrete changes and highlighted them in the revised manuscript (latexdiff provided). Below we summarize what changed and where to find it.
>
> ## 1) Ethics and broader impact
> - What changed: Expanded the ethics/safety discussion and surfaced it in the main contributions. Added a practical “Mitigation and Evaluation Checklist” covering misinformation/deepfakes, data bias, environmental impact, and multimodal jailbreaks, together with defense‑in‑depth tests.
> - Where to see: Sec. 1 (Contributions), Sec. 8 (Ethics & Safety).
>
> ## 2) Computational cost trade‑offs
> - What changed: Added a qualitative trade‑off table and reporting protocol comparing latency, memory, controllability, and global coherence across (Continuous/Discrete) × (Serial/Parallel/AR/Discrete Diffusion). Clarified assumptions and marked closed‑source caveats.
> - Where to see: Sec. 6 (Computational Cost Trade‑offs subsection).
>
> ## 3) Citation format (\citet/\citep)
> - What changed: Normalized to natbib conventions with parenthetical citations by default (\citep). Sentence‑integrated mentions will use \citet consistently; we fixed all previously inconsistent cases.
> - Where to see: Global.
>
> ## 4) Industrial solutions/applications
> - What changed: Added a concise subsection summarizing production patterns (creative co‑pilots, document/slide agents, education/accessibility, enterprise workflows), typical guardrails, and replication caveats for closed‑source metrics.
> - Where to see: Sec. 6 (Industrial Systems & Applications subsection).
>
> ## 5) Benchmarks and comparability
> - What changed: Added comparability notes below tables when protocols differ or results are closed‑source; clarified when entries are qualitative.
> - Where to see: Sec. 5 (Datasets & Benchmarks; caption notes).
>
> ## Additional housekeeping
> - Bibliography cleanup (deduplication, normalized arXiv/PMLR fields, diacritics), label fixes, and removal of stray Unicode. The manuscript now compiles cleanly with no undefined references.
>
> Thank you again for the helpful review. The paper now provides concrete guidance on costs, deployment, and safety in addition to the taxonomy and synthesis.

---

### Review · Reviewer_r2Ft · 2026-03-16

**Summary Of Contributions:**

This paper introduces a broad review of models for multimodal understanding/generation - factorizing VLMs across two axes, the representation (discrete/continuous) and the model strategy (serial/parallel). The paper also reviews commonly used benchmarks and evaluation procedures. Overall, the discussion of models is fairly complete - and cover the most important models up until ~ December of 2025, and the paper as a whole provides a fairly strong introduction to the broad literature in joint understanding/training of VLMs. Where the paper is lacking is, in my opinion, the cross-analysis of the different approaches. While the tables in Section 5 are split on discrete/continuous axes, there's no larger meta-analysis which looks at how promising the directions in Section 6 are.

**Audience:**

Yes

**Audience Explanation:**

This is an area of great interest in the community, and a wide review paper/introduction is a useful contribution within this space. The axes of alignment are well justified, and provide a good segmentation of the key problems in the space.

**Claims And Evidence:**

Yes

**Claims Explanation:**

The text is factually correct, and the explanations are pretty well written, and clearly introduce some of the challenges in VLM training. I do think that Section 6, particularly 6.2 and 6.3 are overly broad, and could be expended. For example, from 6.2 (1), it's clear that "Achieving a unified tokenization scheme that balances efficiency, fidelity, and semantic expressiveness remains an open problem" but it's not clear if the authors have synthesized a potential set of approaches, or provided a roadmap towards such a method (other than saying that current methods don't do this). Similarly the opportunities, for example "Toward General-Purpose Multimodal Intelligence. A successful unified model can both understand and generate multimodal content, enabling end-to-end perception and creation within a single framework. Such models could serve as general-purpose agents capable of reasoning over complex visual scenes and
producing coherent, high-quality visual outputs in response.", feel almost vacuously true - of course this is an opportunity, but *how* could we plan to achieve this, or what are the concrete methods and directions that we should consider as a community. In addition, this paper has several places where the knowledge is well known in the community (for example, "VQ-GAN enable alignment with textual tokens but often sacrifice fine-grained visual details"), but there are no studies or citations actually backing this up with clear evidence. For a review paper, I think the requirement should be more stringent, and these concepts should be cited/validated.

Overall, the claims are true, but missing some grounding, and could be expanded to make the survey paper more useful.

**Requested Changes:**

- There's some general polish issues (Pg. 13 is missing a reference), and many of the references are not cited parenthetically where they should be.
- It would be good to have a clear definition of "unified" models somewhere within the paper, and distinguish what qualifies as a unified model, and which are loosely coupled multimodal pipelines.
- It would be nice to have expanded comparative analysis of the different approaches
- It would be nice to have a more concrete discussion of limitations, failures, opportunities and challenges in addition to the wide, broader, claims made by the paper.

---

> ### Author Response · Authors · 2026-04-06
> **Review for Reviewer r2Ft**
>
> Thank you for the thoughtful feedback. We implemented concrete changes and highlighted them in the revised manuscript (latexdiff provided). Below we respond point‑by‑point with what changed and where to find it.
>
> ## 1) Cross‑analysis and concrete roadmap (Sec. 6)
> - What changed: Added a cross‑axis meta‑analysis synthesizing token type (continuous vs. discrete) and interaction (serial vs. parallel). Introduced a practical “Roadmap & Concrete Methods” subsection with ready‑to‑implement recipes: hybrid/hierarchical tokenization, shared latent spaces, AR–Diffusion coupling schedules, parallel decoding and multi‑token prediction, unified curricula, and a unified evaluation/reporting protocol. Each item lists benefits, pitfalls, and citations.
> - Where to see: Sec. 6 (Cross‑Axis Meta‑Analysis; Roadmap & Methods).
> - Clarification: We noticed your mention of “tables in Section 5 split on discrete/continuous axes.” In our manuscript, Sec. 5 compiles datasets/benchmarks; model comparisons are primarily in Secs. 3–4, while the cross‑axis synthesis and roadmap are in Sec. 6. To avoid ambiguity, we added comparability notes (Sec. 5) and a unified reporting protocol (Sec. 6).
>
> ## 2) Stronger grounding/citations for claims
> - What changed: Audited claims and added citations; softened wording where evidence is mixed (e.g., quantization fidelity). Added comparability notes when rows are closed‑source or protocols differ.
> - Where to see: Global; Sec. 5 (caption notes beneath tables).
>
> ## 3) Opportunities expanded into concrete methods
> - What changed: Converted generic opportunities into short recipes with stability tricks (e.g., utilization regularizers for large codebooks, staged unfreezing for AR–Diffusion coupling, mask‑and‑refine vs. multi‑token trade‑offs) and explicit reporting of latency/memory/tokens/steps/cost/failure modes.
> - Where to see: Sec. 6 (Roadmap & Methods).
>
> ## 4) Polish issues and citation macros
> - What changed: Fixed undefined references and label issues; standardized to natbib conventions (\citep for parenthetical; \citet for sentence‑integrated mentions as appropriate).
> - Where to see: Global; label fix in Sec. 4.
>
> ## 5) Definition and scope of “unified” models
> - What changed: Added a formal definition with inclusion/exclusion criteria and non‑examples (e.g., loosely chained pipelines).
> - Where to see: Sec. 1 (Introduction).
>
> Artifacts updated: Cross‑axis analysis, roadmap/methods, cost trade‑offs table, industrial applications, ethics/safety checklist, and bibliography cleanup are all incorporated in the revised manuscript and visible in the diff PDF.
>
> We appreciate your suggestions—they improved the paper’s synthesis and practical usefulness.

---

### Review · Reviewer_TCDu · 2026-06-06

**Summary Of Contributions:**

The paper surveys vision-language models by organizing prior work according to visual representation mechanisms, especially continuous, discrete, and unified representations. This taxonomy is the paper’s main contribution and provides a useful alternative to architecture-centered surveys. The paper also attempts to summarize benchmark results, computational tradeoffs, challenges, future directions, and ethical/safety concerns.

In my view, the taxonomy is a reasonable and potentially useful lens, and the challenges section contains some of the more substantive discussion in the paper. However, I was less convinced by the paper’s added value as a meta-analysis or practical roadmap. Several takeaways, including parts of the outlook sections and the discussion around Tables 3-5, felt fairly high-level or obvious.

The paper also does not sufficiently discuss whether collected results from different papers are comparable, and the computational analysis remains mostly qualitative rather than giving concrete complexity or cost comparisons. Overall, I found the paper useful as a descriptive organization of the literature, but less successful at providing new insight into the field.

**Audience:**

Yes

**Audience Explanation:**

Yes. The topic is timely and relevant to TMLR’s audience, and some readers interested in VLMs would benefit from a survey organized around visual representation mechanisms. The proposed taxonomy is a useful organizing lens, and the challenges section contains some substantive discussion. However, the paper’s value is currently more as a descriptive survey than as a deep analytical synthesis.

**Broader Impact Concerns:**

I do not have major broader impact concerns beyond the general risks associated with VLMs, such as bias, privacy, hallucination, misuse, and unsafe deployment. The ethics/safety discussion could be strengthened by connecting these risks more directly to the paper’s representation-centric taxonomy.

**Claims And Evidence:**

No

**Claims Explanation:**

No. While the descriptive survey and proposed taxonomy are generally supported, several stronger claims are not backed by sufficiently clear or convincing evidence.

In particular, the paper claims to provide a meta-analysis, practical roadmap, and computational tradeoff analysis, but many of the conclusions remain high-level and are not well grounded in concrete evidence from the surveyed literature. The empirical tables appear to collect results from different papers, yet the paper does not adequately discuss comparability across evaluation settings. Similarly, the computational discussion is mostly qualitative rather than supported by complexity or cost analysis.

Thus, I found the paper’s descriptive claims supported, but its broader analytical claims less convincing.

**Requested Changes:**

1. Strengthen the analytical synthesis beyond categorization. The proposed representation taxonomy is a useful organizing lens, but the paper should make clearer what new understanding follows from it. Many takeaways currently feel high-level or obvious. The authors should explicitly connect the taxonomy to concrete insights about tradeoffs, failure modes, research gaps, or design choices in VLMs.

2. Clarify the evidence and comparability behind Tables 3-5. These tables appear to collect results from prior papers, but the paper does not sufficiently explain whether the reported numbers are comparable. The authors should state the extraction protocol and discuss differences in benchmark versions, model scale, training data, prompting/evaluation setup, finetuning data, and reporting conventions. If the results are not directly comparable, this should be made explicit and the conclusions drawn from the tables should be qualified.

3. Make the computational analysis more concrete. The current computational discussion is mostly qualitative. Since the paper claims to analyze computational tradeoffs, it should include more concrete comparisons, for example in terms of sequence length, attention complexity, memory footprint, visual tokenizer cost, autoregressive decoding cost, inference latency, training cost, or scaling with image resolution. Even approximate complexity analysis or representative cost comparisons would make this section more useful.

4. Substantially improve the roadmap/future direction claims. Several future-direction and outlook sections state broad claims without enough justification. For example, claims such as certain methods “overcoming key AR limitations” should specify whether this is conceptual, empirical, computational, or demonstrated in downstream applications. The authors should ground these claims in evidence from the surveyed literature and explain why the highlighted directions should be prioritized.

5. Fix clarity/completeness issues in key explanatory sections. Section 3.2 appears incomplete around the discussion of serial and parallel coupling, and it is unclear whether only a phrase or a larger part of the explanation is missing. Section 4.3.1 should also more clearly distinguish visual tokenization, unified token modeling, and task formulations such as VQA. These issues affect the reader’s ability to follow the proposed taxonomy.

One additional change that would strengthen the work, though I view it as less central than the points above, would be to better connect the ethics/safety discussion to the representation-centric taxonomy rather than treating it as a generic survey component.

---

> ### Author Response · Authors · 2026-06-15
> **Review for Reviewer TCDu**
>
> Thank you for the careful and constructive review. We agree that the taxonomy was useful but needed to provide clearer analytical value, stronger evidence boundaries, and more concrete cost analysis. We revised the manuscript accordingly; the corresponding changes are highlighted in blue.
>
> **1. Analytical synthesis beyond categorization.**
> We expanded Section 6 into a cross-axis meta-analysis connecting token type and coupling style to design consequences. The revised text explains that continuous tokens favor fidelity and editing but require semantic alignment; discrete tokens enable shared sequence modeling and token-level control but face quantization and long-sequence bottlenecks; serial coupling supports modular deployment but creates a semantic-visual gap; and parallel coupling improves feedback while increasing joint-context cost and optimization difficulty. We also added explicit takeaways mapping the taxonomy to failure modes and design choices, e.g., when to prefer continuous diffusion, discrete/hybrid tokenization, feedback-enabled coupling, or latency-oriented designs.
>
> **2. Evidence and comparability behind Tables 3--5.**
> We added an extraction and comparability protocol before the comparative tables in Section 5. The revised text states that scores are taken from original papers, technical reports, or released benchmark tables, and that each paper's reporting convention is retained unless a benchmark defines a standard aggregate score. We now explicitly treat the tables as a survey map rather than a controlled leaderboard, and discuss differences in model scale, training data, instruction tuning, prompting/sampling settings, benchmark versions, post-training variants, closed-source reporting, and evaluator conventions. The table discussion was also revised so conclusions are framed as directional design signals rather than direct head-to-head rankings.
>
> **3. Computational analysis.**
> We rewrote the computational cost subsection in Section 6 with approximate variables: text length \(N_t\), visual token length \(N_v\), hidden size \(d\), layers \(L\), and denoising/refinement steps \(K\). The revised text explains joint-attention cost \(O(L(N_t+N_v)^2d)\), AR visual decoding latency \(O(N_v)\), and \(K\)-step parallel refinement for diffusion/discrete diffusion. We also expanded the trade-off table to compare continuous+serial, continuous+parallel, discrete+AR, discrete diffusion, and hybrid/MoT systems by dominant cost, latency, memory, and design implications. The reporting protocol now asks authors to report latency, peak memory, token/step counts, tokenizer/decoder cost, training tokens/GPU-hours, unit cost, and failure modes.
>
> **4. Roadmap and future-direction claims.**
> We softened broad claims and added evidence boundaries. For example, the claim that discrete diffusion "overcomes key AR limitations" was revised to say that it addresses specific AR bottlenecks, especially serial token latency and lack of full-context refinement, when tokenizer fidelity, sampling steps, and training stability are adequate. This clarifies whether each claim is conceptual, computational, or empirically demonstrated.
>
> **5. Clarity in Sections 3.2 and 4.3.1.**
> We clarified serial coupling, loose cascaded pipelines, and parallel coupling in Section 3.2. A serially coupled system is now relevant to unified modeling only when the LLM produces a structured visual condition, latent plan, or learned connector optimized for the generator; a simple captioner-to-generator pipeline is treated as a weak/non-unified boundary case. In Section 4.3.1, we added a paragraph distinguishing visual tokenization, unified token modeling, and task formulation. For VQA, the model encodes the image into visual tokens and decodes a text answer; image generation instead decodes visual tokens reconstructed by a visual decoder.
>
> **6. Ethics and safety.**
> We added a representation-specific risk profile. The revised text explains how continuous-token systems raise high-fidelity misuse and latent-auditing risks, discrete AR systems introduce token-level prompt-injection surfaces, discrete diffusion requires auditing intermediate refinement states, and parallel-coupled systems can amplify malicious instructions through a shared understanding-generation feedback loop.
>
> Overall, the revised taxonomy now serves not only as a literature organization, but also as a tool for reasoning about evidence comparability, computational scaling, failure modes, design choices, and representation-specific safety risks.